# Liberty or Depth: Deep Bayesian Neural Nets Do Not Need Complex Weight Posterior Approximations

**Sebastian Farquhar**          **Lewis Smith**          **Yarin Gal**

OATML, Department of Computer Science
University of Oxford
`sebastian.farquhar@cs.ox.ac.uk`

## Abstract

We challenge the longstanding assumption that the mean-field approximation for variational inference in Bayesian neural networks is severely restrictive, and show this is not the case in *deep* networks. We prove several results indicating that deep mean-field variational weight posteriors can induce similar distributions in function-space to those induced by shallower networks with complex weight posteriors. We validate our theoretical contributions empirically, both through examination of the weight posterior using Hamiltonian Monte Carlo in small models and by comparing diagonal- to structured-covariance in large settings. Since complex variational posteriors are often expensive and cumbersome to implement, our results suggest that using mean-field variational inference in a deeper model is both a practical and theoretically justified alternative to structured approximations.

## 1  Introduction

While performing variational inference (VI) in Bayesian neural networks (BNNs) researchers often make the 'mean-field' approximation which assumes that the posterior distribution factorizes over weights (i.e., diagonal weight covariance). Researchers have assumed that using this mean-field approximation in BNNs is a severe limitation. This has motivated extensive exploration of VI methods that explicitly model correlations between weights (see Related Work §2). Furthermore, Foong et al. [2020] have identified pathologies in single-hidden-layer mean-field regression models, and have conjectured that these might exist in deeper models as well.

However, the rejection of mean-field methods comes at a price. Structured covariance methods have worse time complexity (see Table 1) and even efficient implementations take over twice as long to train an epoch as comparable mean-field approaches [Osawa et al., 2019]. Moreover, recent work has succeeded in building mean-field BNNs which perform well (e.g., Wu et al. [2019]), creating a puzzle for those who have assumed that the mean-field approximation is too restrictive.

We argue that for larger, deeper, networks the mean-field approximation matters less to the downstream task of approximating posterior predictive distributions *over functions* than it does in smaller shallow networks. In essence: simple parametric functions need complicated weight-distributions to induce rich distributions in function-space; but complicated parametric functions can induce the same function-space distributions with simple weight-distributions. Complex covariance is computationally expensive and often cumbersome to implement though, while depth can be easy to implement and cheap to compute with standard deep learning packages.

Rather than introducing a new method, we provide empirical and theoretical evidence that some of the widely-held assumptions present in the research community about the strengths and weaknesses of existing methods are incorrect. Even when performing VI in weight-space, one does not care

about the posterior distribution over the weights, $p(\boldsymbol{\theta}|\mathcal{D})$, for its own sake. Most decision processes and performance measures like accuracy or log-likelihood only depend on the induced posterior predictive, the distribution over function values $p(y|\mathbf{x}, \mathcal{D}) = \int p(y|\mathbf{x}, \boldsymbol{\theta})p(\boldsymbol{\theta}|\mathcal{D})d\boldsymbol{\theta}$. One way to have an expressive approximate posterior predictive is to have a simple likelihood function and a rich approximate posterior over the weights, $q(\boldsymbol{\theta})$, to fit to $p(\boldsymbol{\theta}|\mathcal{D})$. But another route to a rich approximate predictive posterior is to have a simple $q(\boldsymbol{\theta})$, and a rich likelihood function, $p(y|\mathbf{x}, \boldsymbol{\theta})$—e.g., a deeper model mapping $\mathbf{x}$ to $y$. These arguments lead us to examine two subtly different hypotheses: one comparing mean-field variational inference (MFVI) to VI with a full- or structured- covariance; and the other comparing the expressive power of a mean-field BNN directly to the true posterior predictive over function values, $p(y|\mathbf{x}, \mathcal{D})$:

**Weight Distribution Hypothesis.** For any BNN with a full-covariance weight distribution, there exists a deeper BNN with a mean-field weight distribution that induces a "similar" posterior predictive distribution in function-space.

**True Posterior Hypothesis.** For any sufficiently deep and wide BNN, and for any posterior predictive, there exists a *mean-field* distribution over the weights of that BNN which induces the same distribution over function values as that induced by the posterior predictive, with arbitrarily small error.

The Weight Distribution Hypothesis would suggest that we can trade a shallow complex-covariance BNN for deeper mean-field BNN without sacrificing the expressiveness of the induced function distribution. We start by analyzing linear models and then use these results to examine deep neural networks with piecewise linear activations (e.g., ReLUs). In linear deep networks—with no non-linear activations—a model with factorized distributions over the weights can be "collapsed" through matrix multiplication into a single *product matrix* with a complex induced distribution. In §3, we analytically derive the covariance between elements of this product matrix. We show that the induced product matrix distribution is very rich, and that three layers of weights suffice to allow non-zero correlations between any pair of product matrix elements. Although we do not show that any full-covariance weight distribution can be represented in this way, we do show that the Matrix Variate Gaussian distribution—a commonly used structured-covariance approximation—is a special case of a three-layer product matrix distribution, allowing MFVI to model rich covariances. In §4 we introduce the *local product matrix*—a novel analytical tool for bridging results from linear networks into deep neural networks with piecewise-linear activations like ReLUs. We apply this more general tool to prove a partial local extension of the results in §3.

The True Posterior Hypothesis states that mean-field weight distributions can approximate the true predictive posterior distribution, and moreover we provide evidence that VI can discover these distributions. In §5 we prove that the True Posterior Hypothesis is true for BNNs with at least two hidden layers, given arbitrarily wide models. In §6.1, we investigate the optima discovered by mean-field VI using Hamiltonian Monte Carlo [Neal, 1995]. We show empirically that even in smaller networks, as the model becomes deeper, we lose progressively less information by using a mean-field approximation rather than full-covariance. We also conduct experiments with deep convolutional architectures and find no significant difference in performance between a particular diagonal and full covariance method (SWAG, [Maddox et al., 2019]), an effect which we find is consistent with results from other papers working with various posterior approximations.

## 2 Related Work

The mean-field approximation has been widely used for variational inference (VI) [Hinton and van Camp, 1993, Graves, 2011, Blundell et al., 2015] (see Appendix A for a brief primer on VI methods for Bayesian neural networks). But researchers have assumed that using the mean-field approximation for Bayesian inference in neural networks is a severe limitation since MacKay [1992] wrote that for BNNs the "diagonal approximation is no good because of the strong posterior correlations in the parameters." Full-covariance VI was there-

| | Complexity | |
|---|---|---|
| | Time | Parameter |
| MFVI [Hinton and van Camp, 1993] | $K^2$ | $K^2$ |
| Full [Barber and Bishop, 1998] | $K^{12}$ | $K^4$ |
| MVG [Louizos and Welling, 2016] | $K^3$ | $K^2$ |
| MVG-Inducing Point [ibid.] | $K^2 + P^3$ | $K^2$ |
| Noisy KFAC [Zhang et al., 2018] | $K^3$ | $K^2$ |

Table 1: Complexity for forward pass in $K$—the number of hidden units for a square weight layer. Mean-field VI has better time complexity and avoids a numerically unstable matrix inversion. Inducing point approximations can help, but inducing dimension $P$ is then a bottleneck.

fore introduced by Barber and Bishop [1998], but it requires many parameters and has poor time complexity. The intractability of full-covariance VI led to extensive research into structured-covariance approximations [Louizos and Welling, 2016, Zhang et al., 2018, Mishkin et al., 2019, Oh et al., 2019]. However, these still have unattractive time complexity compared with mean-field variational inference (MFVI) (see Table 1) and are not widely used. Researchers have also sought to model richer approximate posterior distributions [Jaakkola and Jordan, 1998, Mnih and Gregor, 2014, Rezende and Mohamed, 2015, Louizos and Welling, 2017] or to perform VI directly on the function—but this becomes intractable for high-dimensional input [Sun et al., 2019].

Despite widespread assertions that the mean-field approximation results in a problematically restricted approximate posterior in deep neural networks, there has been no work *in deep neural networks* demonstrating this; theoretical analysis and experimental work supporting this claim is typically based on shallow network architectures. For example, recent work has argued that the mean-field approximation is too restrictive and has identified pathologies in *single-layer* mean-field VI for regression [Foong et al., 2020]. We emphasise their theorems are entirely consistent with ours: we agree that the mean-field approximation could be problematic in small and single-layer models. While Foong et al. [2020] conjecture that the pathologies they identify extend to deeper models and classification problems, they do not prove this (see Appendix B).

In contrast, Hinton and van Camp [1993] hypothesized that even with a mean-field approximating distribution, during optimization the parameters will find a version of the network where this restriction is least costly, which our work bears out. Moreover, others have successfully applied mean-field approximate posteriors [Khan et al., 2018, Osawa et al., 2019, Wu et al., 2019, Farquhar et al., 2020] or even more restrictive approximations [Swiatkowski et al., 2020] in deep models, by identifying and correcting problems like gradient variance that have nothing to do with the restrictiveness of the mean-field approximation.

## 3 Emergence of Complex Covariance in Deep Linear Mean-Field Networks

In this section, we prove that a restricted version of the Weight Distribution Hypothesis is true in linear networks. Although we are most interested in neural networks that have non-linear activations, linear neural networks can be analytically useful [Saxe et al., 2014]. In §4 we give first steps to extend this analysis to non-linear activations.

**Defining a Product Matrix:** Setting the activation function of a neural network, $\phi(\cdot)$, to be the identity turns a neural network into a deep linear model. Without non-linearities the weights of the model just act by matrix multiplication. $L$ weight matrices for a deep linear model can therefore be 'flattened' through matrix multiplication into a single weight matrix which we call the *product matrix*—$M^{(L)}$. For a BNN, the weight distributions induce a distribution over the elements of this product matrix. Because the model is linear, there is a one-to-one mapping between distributions induced over elements of this product matrix and the distribution over linear functions $y = M^{(L)}\mathbf{x}$. This offers us a way to examine exactly which sorts of distributions can be induced by a deep linear model on the elements of a product matrix, and therefore on the resulting function-space.

**Covariance of the Product Matrix:** We derive the analytic form of the covariance of the product matrix in Appendix D.1, explicitly finding the covariance of $M^{(2)}$ and $M^{(3)}$ as well as the update rule for increasing $L$. These results hold for *any* factorized weight distribution with finite first- and second-order moments, not just Gaussian weights. Using these expressions, we show:

**Proposition 1.** *For $L \geq 3$, the product matrix $M^{(L)}$ of factorized weight matrices can have non-zero covariance between any and all pairs of elements. That is, there exists a set of mean-field weight matrices $\{W^{(l)}|1 \leq l < L\}$ such that $M^{(L)} = \prod W^{(l)}$ and the covariance between any possible pair of elements of the product matrix:*

$$\mathrm{Cov}(m_{ab}^{(L)}, m_{cd}^{(L)}) \neq 0, \tag{1}$$

*where $m_{ij}^{(L)}$ are elements of the product matrix in the $i^{th}$ row and $j^{th}$ column, and for any possible indexes $a$, $b$, $c$, and $d$.*

This shows that a deep mean-field linear model is able to induce function-space distributions which would require covariance between weights in a shallower model. This is weaker than the Weight

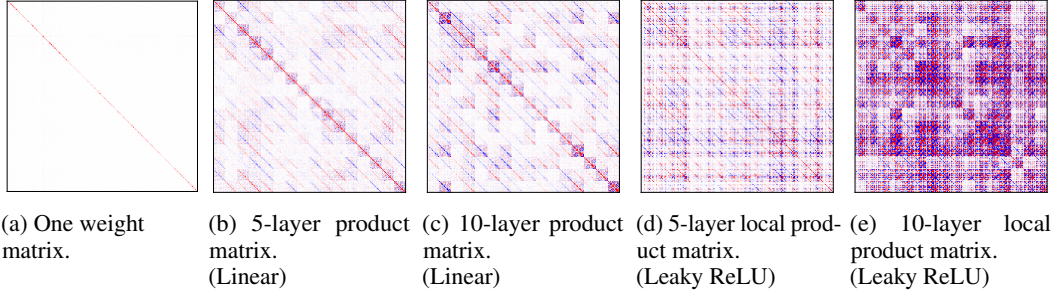

(a) One weight matrix.

(b) 5-layer product matrix. (Linear)

(c) 10-layer product matrix. (Linear)

(d) 5-layer local product matrix. (Leaky ReLU)

(e) 10-layer local product matrix. (Leaky ReLU)

Figure 1: Covariance heatmap for mean-field approximate posteriors trained on FashionMNIST. (a) A single layer has diagonal covariance. (b-c) In a deep linear model the *product matrix* composed of $L$ mean-field weight matrices has off-diagonal covariance induced by the mean-field layers. Redder is more positive, bluer more negative. (d-e) For piecewise non-linear activations we introduce 'local product matrices' (defined in §4) with similar covariance. Shared activations introduce extra correlations. This lets us extend results from linear to piecewise-linear neural networks.

Distribution Hypothesis, because we do not show that all possible fully parameterized covariance matrices between elements of the product matrix can be induced in this way.[1] However, we emphasise that the expressible covariances become very complex. Below, we show that a lower bound on their expressiveness exceeds a commonly used structured-covariance approximate distribution.

**Numerical Simulation:** To build intuition, in Figure 1a–c we visualize the covariance between entries of the product matrix from a deep mean-field VI linear model trained on FashionMNIST. Even though each weight matrix makes the mean-field assumption, the product develops off-diagonal correlations. The experiment is described in more detail in Appendix C.1.

**How Expressive is the Product Matrix?** We show that the Matrix Variate Gaussian (MVG) distribution is a special case of the mean-field product matrix distribution. The MVG distribution is used as a structured-covariance approximation by e.g., Louizos and Welling [2016], Zhang et al. [2018] to approximate the covariance of weight matrices while performing variational inference.[2] We prove in Appendix D.2:

**Proposition 2.** *The Matrix Variate Gaussian (Kronecker-factored) distribution is a special case of the distribution over elements of the product matrix. In particular, for $M^{(3)} = ABC$, $M^{(3)}$ is distributed as an MVG random variable when $A$ and $C$ are deterministic and $B$ has its elements distributed as fully factorized Gaussians with unit variance.*

This directly entails in the linear case that:

**Weak Weight Distribution Hypothesis–MVG.** For any deep linear model with an MVG weight distribution, there exists a deeper linear model with a mean-field weight distribution that induces the same posterior predictive distribution in function-space.

**Remark 1.** *This is a **lower bound** on the expressiveness of the product matrix. We have made very strong restrictions on the parameterization of the weights for the sake of an interpretable result. The unconstrained expressiveness of the product matrix covariance given in Appendix D.1 is much greater. Also note, we do not propose using this in practice, it is purely an analysis tool.*

# 4 Weight Dist. Hypothesis in Deep Piecewise-Linear Mean-Field BNNs

Neural networks use non-linear activations to increase the flexibility of function approximation. On the face of it, these non-linearities make it impossible to consider product matrices. In this section we show how to define the *local product matrix*, which is an extension of the product matrix to widely used neural networks with piecewise-linear activation functions like ReLUs or Leaky ReLUs. For this we draw inspiration from a proof technique by Shamir et al. [2019] which we extend to stochastic matrices. This analytical tool can be used for any stochastic neural network with piecewise linear activations. Here, we use it to extend Lemma 1 to neural networks with piecewise-linear activations.

**Defining a Local Product Matrix:** Neural nets with piecewise-linear activations induce piecewise-linear functions. These piecewise-linear neural network functions define hyperplanes which partition the input domain into regions within which the function is linear. Each region can be identified by a sign vector that indicates which activations are 'switched on'. We show in Appendix D.4.1:

**Lemma 1.** *Consider an input point $\mathbf{x}^* \in \mathcal{D}$. Consider a realization of the model weights $\boldsymbol{\theta}$. Then, for any $\mathbf{x}^*$, the neural network function $f_{\boldsymbol{\theta}}$ is linear over some compact set $\mathcal{A}_{\boldsymbol{\theta}} \subset \mathcal{D}$ containing $\mathbf{x}^*$. Moreover, $\mathcal{A}_{\boldsymbol{\theta}}$ has non-zero measure for almost all $\mathbf{x}^*$ w.r.t. the Lebesgue measure.*

Using a set of $N$ realizations of the weight parameters $\Theta = \{\boldsymbol{\theta}_i \text{ for } 1 \leq i \leq N\}$ we construct a product matrix within $\mathcal{A} = \bigcap_i \mathcal{A}_{\boldsymbol{\theta}_i}$. Since each $f_{\boldsymbol{\theta}_i}$ is linear over $\mathcal{A}$, the activation function can be replaced by a diagonal matrix which multiplies each row of its 'input' by a constant that depends on which activations are 'switched on' (e.g., 0 or 1 for a ReLU). This allows us to compute through matrix multiplication a product matrix of $L$ weight layers $M^{(L)}_{\mathbf{x}^*, \boldsymbol{\theta}_i}$ corresponding to each function realization within $\mathcal{A}$. We construct a local product matrix random variate $P_{\mathbf{x}^*}$, for a given $\mathbf{x}^*$, within $\mathcal{A}$, by sampling these $M^{(L)}_{\mathbf{x}^*, \boldsymbol{\theta}_i}$. The random variate $P_{\mathbf{x}^*}$ is therefore such that $y$ given $\mathbf{x}^*$ has the same distribution as $P_{\mathbf{x}^*} \mathbf{x}^*$ within $\mathcal{A}$. This distribution can be found empirically at a given input point, and resembles the product matrices from linear settings (see Figure 1d–e).

**Covariance of the Local Product Matrix:** We can examine this local product matrix in order to investigate the covariance between its elements. We prove in D.4 that:

**Proposition 3.** *Given a mean-field distribution over the weights of neural network $f$ with piecewise linear activations, $f$ can be written in terms of the local product matrix $P_{\mathbf{x}^*}$ within $\mathcal{A}$.*

*For $L \geq 3$, for activation functions which are non-zero everywhere, there exists a set of weight matrices $\{W^{(l)} | 1 \leq l < L\}$ such that all elements of the local product matrix have non-zero off-diagonal covariance:*

$$\text{Cov}(p^{\mathbf{x}^*}_{ab}, p^{\mathbf{x}^*}_{cd}) \neq 0, \tag{2}$$

*where $p^{\mathbf{x}^*}_{ij}$ is the element at the $i^{th}$ row and $j^{th}$ column of $P_{\mathbf{x}^*}$.*

Proposition 3 is weaker than the Weight Distribution Hypothesis. Once more, we do not show that all full-covariance weight distributions can be exactly replicated by a deeper factorized network. We now have non-linear networks which give richer functions, potentially allowing richer covariance, but the non-linearities have introduced analytical complications. However, it illustrates the way in which deep factorized networks can emulate rich covariance in a shallower network.

**Remark 2.** *Proposition 3 is restricted to activations that are non-zero everywhere although we believe that in practice it will hold for activations that can be zero, like ReLU. If the activation can be zero then, for some $\mathbf{x}^*$, enough activations could be 'switched off' such that the effective depth is less than three. This seems unlikely in a trained network, since it amounts to throwing away most of the network's capacity, but we cannot rule it out theoretically. In Appendix D.4 we empirically corroborate that activations are rarely all 'switched off' in multiple entire layers.*

**Numerical Simulation:** We confirm empirically that the local product matrix develops complex off-diagonal correlations using a neural networkwith Leaky ReLU activations trained on FashionMNIST using mean-field variational inference. We estimate the covariance matrix using 10,000 samples of a trained model (Figure 1d–e). Just like in the linear case (Figure 1a–c), as the model gets deeper the induced distribution on the product matrix shows complex off-diagonal covariance. There are additional correlations between elements of the product matrix based on which activation pattern is predominantly present at that point in input-space. See Appendix C.1 for further experimental details.

# 5 True Posterior Hypothesis in Two-Hidden-Layer Mean-Field Networks

We prove the True Posterior Hypothesis using the universal approximation theorem (UAT) due to Leshno et al. [1993] in a stochastic adaptation by Foong et al. [2020]. This shows that a BNN with a mean-field approximate posterior with at least two layers of hidden units can induce a function-space distribution that matches any true posterior distribution over function values arbitrarily closely, given arbitrary width. Our proof formalizes and extends a remark by Gal [2016, p23] concerning multi-modal posterior predictive distributions.

**Proposition 4.** *Let $p(\mathbf{y} = \mathbf{Y}|\mathbf{x}, \mathcal{D})$ be the probability density function for the posterior predictive distribution of any given multivariate regression function, with $\mathbf{x} \in \mathcal{A}$ where $\mathcal{A}$ is a compact set in $\mathbb{R}^D$, $\mathbf{y} \in \mathbb{R}^K$, and $\mathbf{Y}$ the posterior predictive random variable. Let $f(\cdot)$ be a Bayesian neural network with two hidden layers. Let $\hat{\mathbf{Y}}$ be the random vector defined by $f(\mathbf{x})$. Then, for any $\epsilon, \delta > 0$, there exists a set of parameters defining the neural network $f$ such that the absolute value of the difference in probability densities for any point is bounded:*

$$\forall \mathbf{y}, \mathbf{x} \in \mathcal{A}, i : \quad \Pr\left(|p(y_i = \hat{Y}_i) - p(y_i = Y_i|\mathbf{x}, \mathcal{D})| > \epsilon\right) < \delta, \tag{3}$$

*so long as: the activations of $f$ are non-polynomial, non-periodic, and have only zero-measure non-monotonic regions, the first hidden layer has at least $D + 1$ units, the second hidden layer has an arbitrarily large number of units, the cumulative density function of the posterior predictive is continuous in output-space, and the probability density function is continous and finite non-zero everywhere. Here, the probability bound is with respect to the distribution over a subset of the weights described in the proof, $\boldsymbol{\theta}_{\mathrm{Pr}}$, while one weight distribution $\boldsymbol{\theta}_Z$ induces the random variable $\hat{\mathbf{Y}}$.*

The full proof is provided in Appendix D.5. Intuitively, we define a $q(\boldsymbol{\theta})$ to induce an arbitrary distribution over hidden units in the first layer and using the remaining weights and hidden layer we approximate the inverse cumulative density function of the true posterior predictive by the UAT. It follows from Proposition 4 that it is possible, in principle, to learn a mean-field approximate posterior which induces the true posterior distribution over predictive functions. Our proof strengthens a result by Foong et al. [2020] which considers only the first two moments of the posterior predictive.

There are important limitations to this argument to bear in mind. First, the UAT might require arbitrarily wide models. Second, to achieve arbitrarily small error $\delta$ it is necessary to reduce the weight variance. Both of these might result in very low weight-space evidence lower-bounds (ELBOs). Third it may be difficult in practice to choose a prior in weight-space that induces the desired prior in function space. Fourth, although the distribution in weight space that maximizes the marginal likelihood will also maximize the marginal likelihood in function-space within that model class, the same is not true of the weight-space ELBO and functional ELBO. Our proposition therefore does not show that such an approximate posterior will be found by VI. We investigate this empirically below.

# 6 Empirical Validation

We have already considered the Weight Distribution Hypothesis theoretically and confirmed through numerical simulation that deeper factorized networks can show off-diagonal covariance in the product matrix. We now examine the True Posterior Hypothesis empirically from two directions. First, we examine the true posterior of the weight distribution using Hamiltonian Monte Carlo and consider the extent to which the mean-field assumption makes it harder to match a mode of the true posterior. Second, we work off the assumption that a worse approximation of the true posterior would result in significantly worse performance on downstream tasks, and we show that we are unable to find such a difference in both large- and small-scale settings.

## 6.1 Examining the True Posterior with Hamiltonian Monte Carlo

Proposition 4 proves the True Posterior Hypothesis in sufficiently wide models of two or more layers. Here, we examine the true posterior distribution using Hamiltonian Monte Carlo (HMC) and show that even in narrow deep models there are modes of the true posterior that are approximately mean-field. We examine a truly full-covariance posterior, not even assuming that layers are independent of each other, unlike Barber and Bishop [1998] and most structured covariance approximations.

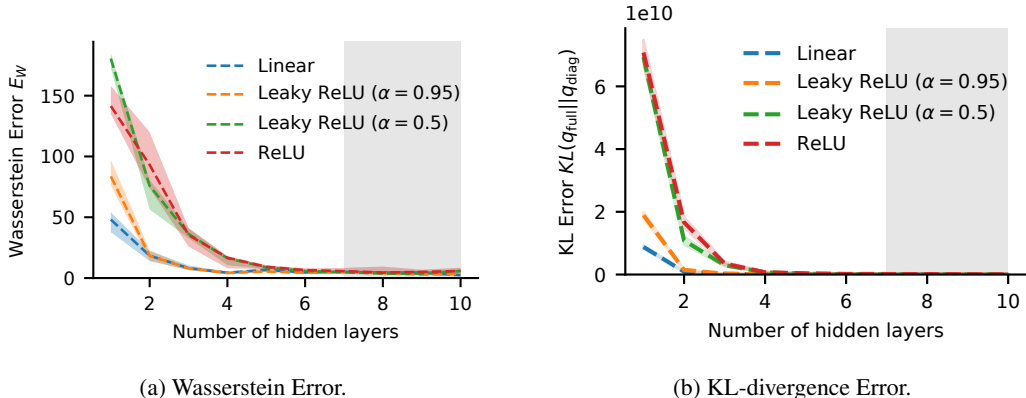

(a) Wasserstein Error.

(b) KL-divergence Error.

Figure 2: For all activations and both error measures, large error in shallow networks almost disappears with depth. All models have ~1,000 parameters. Shaded depths: less reliable HMC samples.

**Methodology:** To approximate the true posterior distribution—$p(\boldsymbol{\theta})$—we use the No-U-turn HMC sampling method [Hoffman and Gelman, 2014]. We then aim to determine how much worse a mean-field approximation to these samples is than a full-covariance one. To do this, we fit a full-covariance Gaussian distribution to the samples from the true posterior—$\hat{q}_{\text{full}}(\boldsymbol{\theta})$—and a Gaussian constrained to be fully-factorized—$\hat{q}_{\text{diag}}(\boldsymbol{\theta})$. We deliberately do not aim to sample from multiple modes of the true posterior—VI is an essentially mode-seeking method.[3] Each point on the graph represents an average over 20 restarts (over 2.5 million model evaluations per point on the plot). For all depths, we adjust the width so that the model has roughly 1,000 parameters and train on the binary classification 'two-moons' task. We report the sample test accuracies and acceptance rates in Appendix C.3 and provide a full description of the method. We consider two measures of distance:

1. **Wasserstein Distance:** We estimate the $L_2$-Wasserstein distance between samples from the true posterior and each Gaussian approximation. Define the Wasserstein Error:

$$E_W = W(p(\boldsymbol{\theta}), \hat{q}_{\text{diag}}(\boldsymbol{\theta})) - W(p(\boldsymbol{\theta}), \hat{q}_{\text{full}}(\boldsymbol{\theta})). \tag{4}$$

   If the true posterior is fully factorized, then $E_W = 0$. The more harmful a fully-factorized assumption is to the approximate posterior, the larger $E_W$ will be.

2. **KL-divergence:** We estimate the KL-divergence between the two Gaussian approximations. Define the KL Error:

$$E_{KL} = KL\big(\hat{q}_{\text{full}}(\boldsymbol{\theta}) \,\|\, \hat{q}_{\text{diag}}(\boldsymbol{\theta})\big). \tag{5}$$

   This represents a worst-case information loss from using the diagonal Gaussian approximation rather than a full-covariance Gaussian, measured in nats (strictly, the infimum information loss under any possible discretization [Gray, 2011]). $E_{KL} = 0$ when the mode is naturally diagonal, and is larger the worse the approximation is. (Note that we do not directly involve $p(\boldsymbol{\theta})$ here because KL-divergence does not follow the triangle inequality.)

Note that we are only trying to establish how costly the *mean-field* approximation is relative to full covariance, not how costly the *Gaussian* approximation is.

**Results:** In Figure 2a we find that the Wasserstein Error introduced by the mean-field approximation is large in shallow networks but falls rapidly as the models become deeper. In Figure 2b we similarly show that the KL-divergence Error is large for shallow networks but rapidly decreases. Although these models are small, this is very direct evidence that there are mean-field modes of the true posterior of a deeper Bayesian neural network.

In all cases, this is true regardless of the activation we consider, or whether we use any activation at all. Indeed we find that a non-linear model with a very shallow non-linearity (LeakyReLU with $\alpha = 0.95$) behaves very much like a deep linear model, while one with a sharper but still shallow non-linearity ($\alpha = 0.5$) behaves much like a ReLU. This suggests that the shape of the true posterior modes varies

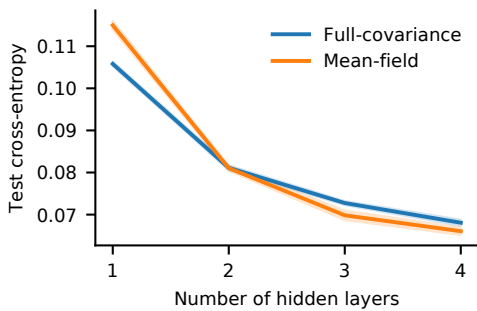 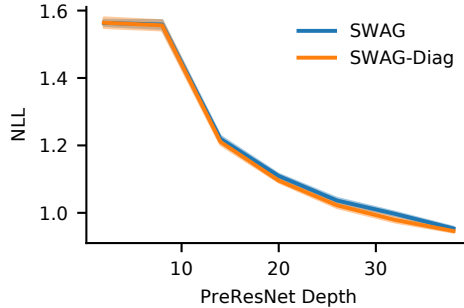

Figure 3: Full- vs diagonal-covariance. After two hidden layers mean-field matches full-covariance. Iris dataset.

Figure 4: CIFAR-100. Diagonal- and Structured-SWAG show similar log-likelihood in PresNets of varying depth.

between linear models and non-linear ones more as a matter of degree than kind, suggesting that our analysis in §3 has bearing on the non-linear case.

## 6.2 Factorization and Downstream Task Performance

Here, we compare the performance of Bayesian neural networks with complex posterior approximations to those with mean-field approximations. We show that over a spectrum of model sizes, performance does not seem to be greatly determined by the approximation.

**Depth in Full- and Diagonal-covariance Variational Inference.** Training with full-covariance variational inference is intractable, except for very small models, because of optimization difficulties. In Figure 3, we show the test cross-entropy of small models of varying depths on the Iris dataset from the UCI repository. With one layer of hidden units the full-covariance posterior achieves lower cross-entropy. For deeper models, however, the mean field network matches the full-covariance one. Full details of the experiment can be found in Appendix C.2.

**Structured- and Diagonal-covariance Uncertainty on CIFAR-100.** Although we cannot compute samples from the true posterior in larger models, we attempt an approximate investigation using SWAG [Maddox et al., 2019]. This involves fitting a Gaussian distribution to approximate Stochastic Gradient-Markov Chain Monte Carlo samples on CIFAR-100. SWAG approximates the Gaussian distribution with a low rank empirical covariance matrix, while SWAG-Diag uses a full-factorized Gaussian. The resulting distributions are some indication of large-model posterior behaviour, but cannot carry too much weight. We show in Figure 4 that there is no observable difference in negative log-likelihood between the diagonal and low-rank approximation (or accuracy, see Appendix C.4). All of the models considered have more than two layers of hidden units (the minimum size of a PresNet). This suggests that there is a mode of the true posterior over weights for these deeper models that is sufficiently mean-field that a structured approximation provides little or no benefit. It also suggests that past a threshold of two hidden layers, further depth is not essential.

**Large-model Mean-field Approximations on Imagenet.** The performance of mean-field and structured-covariance methods on large-scale tasks can give some sense of how restrictive the mean-field approximation is. Mean-field methods have been shown to perform comparably to structured methods in large scale settings like Imagenet, both in accuracy and measures of uncertainty like log-likelihood and expected calibration error (ECE) (see Table 2). For VOGN [Osawa et al., 2019] which explicitly optimizes for a mean-field variational posterior, the mean-field model is marginally better in all three measures. For SWAG, the accuracy is marginally better and log-likelihood and ECE marginally worse for the diagonal approximation. This is consistent with the idea that there are some modes of large models that are approximately mean-field (which VOGN searches for but SWAG does not) but that not all modes are. These findings offer some evidence that the importance of structured covariance is at least greatly diminished in large-scale models, and may not be worth the additional computational expense and modelling complexity. A table with standard deviations and comparison for CIFAR-10 is provided in Appendix C.5.

| Architecture | Method | Covariance | Accuracy | NLL | ECE |
|---|---|---|---|---|---|
| ResNet-18 | VOGN[‡] | Diagonal | 67.4% | 1.37 | 0.029 |
| ResNet-18 | Noisy K-FAC[††] | MVG | 66.4% | 1.44 | 0.080 |
| DenseNet-161 | SWAG-Diag[†] | Diagonal | 78.6% | 0.86 | 0.046 |
| DenseNet-161 | SWAG[†] | Low-rank | 78.6% | 0.83 | 0.020 |
| ResNet-152 | SWAG-Diag[†] | Diagonal | 80.0% | 0.86 | 0.057 |
| ResNet-152 | SWAG[†] | Low-rank | 79.1% | 0.82 | 0.028 |

Table 2: Imagenet diagonal- and structure-covariance methods. Both approximations have similar accuracies, log-likelihoods, and expected calibration errors. Suggests that covariance matters less in large models, as predicted. [†] [Maddox et al., 2019]. [‡] [Osawa et al., 2019]. [††] [Zhang et al., 2018] as reported by Osawa et al. [2019].

## 7 Discussion

We have argued that deeper models with mean-field approximate posteriors can act like shallower models with much richer approximate posteriors. In deep linear models, a product matrix with rich covariance structure is induced by mean-field approximate posterior distributions—in fact, the Matrix Variate Gaussian is a special case of this induced distribution for at least three weight layers (two layers of hidden units) (§3). We provided a new analytical tool to extend results from linear models to piecewise linear neural networks (e.g., ReLU activations): the local product matrix. In addition, examination of the induced covariance in the local product matrix (§4) and posterior samples with HMC (§6.1) suggest that the linear results are informative about non-linear networks.

Moreover, we have proved that neural networks with at least two layers of hidden units and mean-field weight distributions can approximate any posterior distribution over predictive functions. In sufficiently deep models, the performance gap between mean-field and structured-covariance approximate posteriors becomes small or non-existent, suggesting that modes of the true posterior in large settings may be approximately mean-field.

Our work challenges the previously unchallenged assumption that mean-field VI fails because the posterior approximation is too restrictive. Instead, rich posterior approximations and deep architectures are complementary ways to create rich approximate posterior distributions over predictive functions. So long as a network has at least two layers of hidden units, increasing the parameterization of the neural network allows some modes of the true posterior over weights to become approximately mean-field. This means that approximating posterior functions becomes easier for mean-field variational inference in larger models—making it more important to address other challenges for MFVI at scale.

## Acknowledgements

We would especially like to thank Adam Cobb for his help applying the Hamiltorch package [Cobb et al., 2019]; Angelos Filos for his help with the experiment in Figure 5; and Andrew Foong, David Burt, and Yingzhen Li for their conversations regarding the proof of Proposition 4. The authors would further like to thank for their comments and conversations Wendelin Boehmer, David Burt, Adam Cobb, Gregory Farquhar, Andrew Foong, Raza Habib, Andreas Kirsch, Yingzhen Li, Clare Lyle, Michael Hutchinson, Sebastian Ober, Hippolyt Ritter, and Jan Wasilewski as well as anonymous reviewers who have generously contributed their time and expertise.

## Broader Impact

Our work addresses a growing need for scalable neural network systems that are able to express sensible uncertainty. Sensible uncertainty is essential in systems that make important decisions in production settings. Despite that, the most performant production systems often rely on large deterministic deep learning models. Historically, uncertainty methods have often prioritized smaller settings where more theoretically rigorous methods could be applied. Our work demonstrates the theoretical applicability of cheap uncertainty approximation methods that do not attempt to model complex correlations between weight distributions in those large-scale settings. This resolves

something the field has assumed is a tension between good uncertainty and powerful models—we show that some modes of the variational weight posterior might be closer to mean-field in bigger models. So using a bigger model causes more restrictive approximation methods to become more accurate.

In principle, this could allow Bayesian neural networks with robust uncertainty to be deployed in a wide range of settings. If we are right, this would be a very good thing. The main downside risk of our research is that if we are wrong, and people deploy these systems and incorrectly rely on their uncertainty measures, then this could result in accidents caused by overconfidence. We therefore recommend being extremely cautious in how a business or administrative decision process depends on any uncertainty measures in critical settings, as is already good practice for non-uncertainty-aware decisions and in non-neural network uncertain machine learning systems.

## Funding Disclosure

This work was supported by the EPSCRC CDTs for Cyber Security and for Autonomous Intelligent Machines and Systems at the University of Oxford. It was also supported by the Alan Turing Institute.

## Footnotes

[1]E.g., a full-covariance layer has more degrees of freedom than a three-layer mean-field product matrix (one of the weaknesses of full-covariance in practice). An $L$-layer product matrix of $K \times K$ Gaussian weight matrices has $2LK^2$ parameters, but one full-covariance weight matrix has $K^2$ mean parameters and $K^2(K^2 + 1)/2$ covariance parameters. Note also that the distributions over the elements of a product matrix composed of Gaussian layers are not in general Gaussian (see Appendix D.3 for more discussion of this point).

[2]In some settings, MVG distributions can be indicated by the Kronecker-factored or K-FAC approximation. In MVGs, the covariance between elements of an $n_0 \times n_1$ weight matrix can be described as $\Sigma = V \otimes U$ where $U$ and $V$ are positive definite real scale matrices of shape $n_0 \times n_0$ and $n_1 \times n_1$.

[3]In fact, in order to ensure that we do not accidentally capture multiple modes, we use the dominant mode of a mixture of Gaussians model selected using the Bayesian Information Criterion (see Appendix C.3 for details).

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
