[Supplementary Material]

# A   Introduction to Variational Inference in Bayesian Neural Networks

A Bayesian neural network (BNN) places a distribution over the weights of a neural network [MacKay, 1992]. We hope to infer the posterior distribution over the weights given the data, $p(\boldsymbol{\theta}|\mathcal{D})$—although ultimately we are interested in a posterior distribution over functions, as described by Sun et al. [2019]. Because this is intractable, we seek an approximate posterior $q(\boldsymbol{\theta})$ to be as close as possible to the posterior over the weights. Variational inference (VI) is one method for estimating that approximate posterior, in which we pick an approximating distribution and minimize the KL-divergence between it and the true posterior. This KL divergence is the Evidence Lower Bound (ELBO), expressed using the prior over the weight distributions $p(\boldsymbol{\theta})$. We therefore minimize the negative ELBO:

$$\mathcal{L}_{MFVI} = \overbrace{KL\big(q(\boldsymbol{\theta}) \parallel p(\boldsymbol{\theta})\big)}^{\text{prior regularization}} - \overbrace{\mathbb{E}\big[\log p(y|\mathbf{x}, \boldsymbol{\theta})\big]}^{\text{data likelihood}}. \tag{6}$$

For the full-covariance Gaussian approximate posterior [Barber and Bishop, 1998], the model weights for each layer $\boldsymbol{\theta}_i$ are distributed according to the multivariate Gaussian distribution $\mathcal{N}(\boldsymbol{\mu}_i, \Sigma_i)$. This is already a slight approximation, as it assumes layers are independent of each other. This is important for our analysis in §3 and §4, though note that the experiment in §6.1 does not assume any independence between layers.

The mean-field approximation restricts $\Sigma_i$ to be a diagonal covariance matrix, or equivalently assumes that the probability distribution can be expressed as a product of individual weight distributions:

$$\mathcal{N}(\boldsymbol{\mu}_i, \Sigma_i) = \prod_j \mathcal{N}(\mu_{ij}, \sigma_{ij}). \tag{7}$$

The mean-field approximation greatly reduces the computational cost of both forward and backwards propagation, and reduces the number of parameters required to store the model from order $n^2$ in the number of weights to order $n$. The implementation of mean-field variational inference which we use is based on Blundell et al. [2015], who show how to use a stochastic estimator of the ELBO.

# B   Discussions of Foong et al. 2020

Our work discusses some similar topics to those discussed by Foong et al. [2020], in work which was developed in parallel to this paper. In particular, they reach a different conclusion as to the value of mean-field variational inference in *deep* BNNs. We find their work insightful, but we think it is important to be very precise about where their results do and do not apply, in order to best understand their implications. Here, we briefly describe several of their main results, emphasising that our analysis is not in conflict with any of their proofs.

Empirically, they focus on the posterior distribution over function outputs in small models, where they find that the learned function distributions with mean-field variational inference for regression tend to be overconfident, even in slightly deeper models. We also find that in relatively small regression tasks MFVI does not perform particularly well, which aligns with their results. However, below, we note that in fact their theoretical results suggest that MFVI may have problems with regression that do not extend to classification tasks. Moreover, it is important to note that the largest of their experiments considers data with only 16-dimensional inputs, with only 55 training points and very small models with 4 layers of 50 hidden units. In contrast, our work analyses much larger models and datasets, where it is admittedly harder to compare to a reference posterior in function-space. However, these are the situations where MFVI would be more typical for deep learning. We therefore feel that, while there is significant room for further investigation, on the balance of the current evidence MFVI seems quite appropriate for large-scale deep learning especially in classification tasks.

Foong et al. [2020] base much of their interpretation on the inability of single-layer mean-field networks to have appropriate 'in-between' uncertainty. That is, they observe that if a model is trained on data in two regions of input space which are separated, it ought to be able to be significantly more uncertain between those two regions. They prove their Theorem 1 which states that the variance of the function expressed by a single-layer mean-field network between two points cannot be greater than the sum of the variances of the function at those points, subject to a number of very important caveats. Our work is largely concerned with models with more than a single hidden layer, where

their theorem does not apply. Nevertheless, Foong et al. [2020] hypothesize the pathologies that they identify might extend to deeper settings, so we note some further limitations of their proof.

1. Theorem 1 applies to single- or multi-output regression models, and to the individual logits in classification models, but makes no predictions about the variance of classification *decisions* (because this depends on the variance of the argmax of the logits).

2. Theorem 1 is strongest in a 1-dimensional input space. In higher dimensions, Foong et al. [2020] show as a corollary that the 'in-between' variance is bounded by the sum of the variances of the hypercube of points including that space. But the number of such points grows exponentially with the dimensionality, meaning that in even only 10 dimensions, the in-between variance could be as much as 1024 times greater than the average edge variance. This means that in high-dimensional input spaces, this bound can be extremely loose and does not neccesarily preclude even single layer models from having significant 'in between' uncertainty.

3. The line-segments where Theorem 1 applies are not fully general. They must either go through the origin, or be orthogonal to one of the input basis vectors and cross a projection of the origin. This makes their result sensitive to translation and rotation of the input space. However, the authors do provide some empirical evidence that 'in between' uncertainty is too low on more general lines in input space for small models.

They also prove their Theorem 3, which establishes that deeper mean-field networks do not have the pathologies that apply in the limited single-layer regression settings identified in Theorem 1. We consider a similar result (Proposition 4) which is more general because it considers more than the first two moments of the distribution. Unlike Foong et al. [2020] we see this result as potentially promising that deep mean-field networks can be very expressive. This is perhaps because their empirical results suggest that deeper mean-field networks have poor performance in practice. This may be because their experiments mostly focus on low-dimensional data with small numbers of datapoints and comparatively small networks, while we consider the larger settings.

We note also that we find that deeper networks are able to show in-between uncertainty in regression, if not necessarily capture it as fully as something like HMC. In Figure 5 we show how increasing depth increases the ability of MFVI neural networks to capture in-between uncertainty even on low-dimensional data.

Here, each layer has 100 hidden units trained using mean-field variational inference on a sythetic dataset in order to demonstrate the possibility of 'in-between' uncertainty. Full experimental settings are provided in Table 3. The toy function used is $y = \sin(4(x - 4.3)) + \epsilon$ where $\epsilon \sim \mathcal{N}(0, 0.05^2)$. We sample 750 points in the interval $-2 \leq x \leq -1.4$ and another 750 points in the interval $1.0 \leq x \leq 1.8$. We considered a range of temperatures between 0.1 and 100 in order to select the right balance between prior and data. Note of course that while our figure in the main body suffices to demonstrate the existence claim that there are deep networks that perform well, of course a single case of a one-layer network performing badly does not show that all one-layer networks perform badly.

## C Experimental Details

### C.1 Full Description of Covariance Visualization

Here we provide details on the method used to produce Figure 1. The linear version of the visualization is discussed in §3 and the piecewise-linear version is discussed in §4.

In all cases, we train a neural network using mean-field variational inference in order to visualize the covariance of the product matrix. The details of training are provided in Table 4. The product matrix is calculated from the weight matrices of an $L$-layer network. In the linear case, this is just the matrix product of the $L$ layers. In the piecewise-linear case the definition of the product matrix is described in more detail in Appendix D.4.2. All covariances are calculated using 10,000 samples from the converged approximate posterior. Note that for $L$ weight matrtices there are $L - 1$ layers of hidden units.

We compare these learned product matrices, in Figure 6, to a randomly sampled product matrix. To do so, we sample weight layers whose entries are distributed normally. Each weight is sampled with

(a) 1-layer BNN

(b) 3-layer BNN

Figure 5: In-between uncertainty. With a single hidden layer, mean-field variational inference cannot capture in-between uncertainty at all in low-dimensional settings. But by adding more layers, we increase the ability to capture in-between uncertainty, though still not to the point of HMC. Note, of course, that neural networks are at their most useful with higher dimensional data where in-between uncertainty results are exponentially weaker.

| Hyperparameter | Setting description |
|---|---|
| Architecture | MLP |
| Number of hidden layers | 3 |
| Layer Width | 100 |
| Activation | Leaky ReLU |
| Approximate Inference Algorithm | Mean-field VI (Flipout [Wen et al., 2018]) |
| Optimization algorithm | Amsgrad [Reddi et al., 2018] |
| Learning rate | $10^{-3}$ |
| Batch size | 250 |
| Variational training samples | 1 |
| Variational test samples | 1 |
| Temperature | 65 |
| Noise scale | 0.05 |
| Epochs | 6000 |
| Variational Posterior Initialization | Tensorflow Probability default |
| Prior | $\mathcal{N}(0, 1.0^2)$ |
| Dataset | Toy (see text) |
| Number of training runs | 1 |
| Number of evaluation runs | 1 |
| Measures of central tendency | n.a. |
| Runtime per result | $< 5m$ |
| Computing Infrastructure | Nvidia GTX 1060 |

Table 3: Experimental Setting—Toy Regression Visualization. Note that for these visualizations we are purely demonstrating the possibility of in-between uncertainty. As a result, a single training/evaluation run suffices to make an existence claim, so we do not do multiple runs in order to calculate a measure of central tendency.

Figure 6: Unlike the covariance matrices of product matrix entries in models trained on real data, a randomly sampled product matrix does not show obvious block structure, though the noise makes it hard to be sure. This model has 5 linear layers.

Figure 7: The local product matrix also allows the unimodal Gaussian layers to approximate multimodal distributions over product matrix entries. Here, we show an example density over a product matrix element, from a three-layer Leaky ReLU model with mean-field Gaussian distributions over each weight trained on FashionMNIST.

| Hyperparameter | Setting description |
|---|---|
| Architecture | MLP |
| Number of hidden layers | 0-9 |
| Layer Width | 16 |
| Activation | Linear or Leaky Relu with $\alpha = 0.1$ |
| Approximate Inference Algorithm | Mean-field Variational Inference |
| Optimization algorithm | Amsgrad [Reddi et al., 2018] |
| Learning rate | $10^{-3}$ |
| Batch size | 64 |
| Variational training samples | 16 |
| Variational test samples | 16 |
| Epochs | 10 |
| Variational Posterior Initial Mean | He et al. [2016] |
| Variational Posterior Initial Standard Deviation | $\log[1 + e^{-3}]$ |
| Prior | $\mathcal{N}(0, 0.23^2)$ |
| Dataset | FashionMNIST [Xiao et al., 2017] |
| Preprocessing | Data normalized $\mu = 0$, $\sigma = 1$ |
| Validation Split | 90% train - 10% validation |
| Number of training runs | 1 |
| Number of evaluation runs | 1 |
| Measures of central tendency | n.a. |
| Runtime per result | $< 3m$ |
| Computing Infrastructure | Nvidia GTX 1080 |

Table 4: Experimental Setting—Covariance Visualization. Note that for these visualizations we are purely demonstrating the possibility of off-diagonal covariance. As a result, a single training/evaluation run suffices to make an existence claim, so we do not do multiple runs in order to calculate a measure of central tendency.

| Hyperparameter | Setting description |
| --- | --- |
| Architecture | MLP |
| Number of hidden layers | 1-4 |
| Layer Width | 4 |
| Activation | Leaky Relu |
| Approximate Inference Algorithm | Variational Inference |
| Optimization algorithm | Amsgrad [Reddi et al., 2018] |
| Learning rate | $10^{-3}$ |
| Batch size | 16 |
| Variational training samples | 1 |
| Variational test samples | 1 |
| Epochs | 1000 (early stopping patience=30) |
| Variational Posterior Initial Mean | He et al. [2016] |
| Variational Posterior Initial Standard Deviation | $\log[1 + e^{-6}]$ |
| Prior | $\mathcal{N}(0, 1.0^2)$ |
| Dataset | Iris [Xiao et al., 2017] |
| Preprocessing | None. |
| Validation Split | 100 train - 50 test |
| Number of training runs | 100 |
| Number of evaluation runs | 100 |
| Measures of central tendency | (See text.) |
| Runtime per result | $< 5m$ |
| Computing Infrastructure | Nvidia GTX 1080 |

Table 5: Experimental Setting—Full Covariance.

standard deviation 0.3 and with a mean 0.01 and each weight matrix is 16x16. This visualization is with a linear product matrix of 5 layers.

Further, since researchers often critique a Gaussian approximate posterior because it is unimodal, we confirm empirically that multiple mean-field layers can induce a multi-modal product matrix distribution. In Figure 7 we show a density over an element of the local product matrix from three layers of weights in a Leaky ReLU BNN with $\alpha = 0.1$. The induced distribution is multi-modal. We visually examined the distributions over 20 randomly chosen entries of this product matrix and found that 12 were multi-modal. We found that without the non-linear activation, none of the product matrix entry distributions examined were multimodal, suggesting that the non-linearities in fact play an important role in inducing rich predictive distributions by creating modes corresponding to activated sign patterns.

## C.2 Effect of Depth Measured on Iris Experimental Settings

We describe the full- and diagonal-covariance experiment settings in Table 5. We use a very small model on a small dataset because full-covariance variational inference is unstable, requiring a matrix inversion of a $K^4$ matrix for hidden unit width $K$. Unfortunately, for deeper models the initializations still resulted in failed training for some seeds. To avoid this issue, we selected the 10 best seeds out of 100 training runs, and report the mean and standard error for these. Because we treat full- and diagonal-covariance in the same way, the resulting graph is a fair reflection of their relative best-case merits, but not intended as anything resembling a 'real-world' performance benchmark.

Readers may consider the Iris dataset to be unhelpfully small, however this was a necessary choice. We note that the small number of training points creates a broad posterior, which is the best-case scenario for a full-covariance approximate posterior.

## C.3 HMC Experimental Settings

We begin by sampling from the true posterior using HMC.

| # Hidden Layers | ReLU | | Leaky ReLU 0.5 | | Leaky ReLU 0.95 | | Linear | |
|---|---|---|---|---|---|---|---|---|
| | Test Acc. | Acceptance | Test Acc. | Acceptance | Test Acc. | Acceptance | Test Acc. | Acceptance |
| 1 | 99.1% | 84.8% | 98.4% | 84.9% | 91.9% | 85.4% | 83.9% | 78.0% |
| 2 | 99.7% | 77.0% | 99.5% | 73.9% | 96.4% | 76.3% | 84.2% | 44.4% |
| 3 | 99.1% | 58.0% | 99.6% | 46.3% | 97.2% | 74.5% | 84.4% | 37.0% |
| 4 | 99.5% | 62.2% | 99.6% | 50.9% | 95.8% | 68.2% | 84.4% | 43.2% |
| 5 | 98.1% | 61.8% | 99.5% | 53.8% | 98.4% | 62.4% | 84.3% | 35.2% |
| 6 | 95.4% | 78.5% | 99.6% | 51.0% | 98.0% | 62.6% | 84.1% | 33.7% |
| 7 | 92.7% | 68.1% | 99.7% | 54.6% | 97.5% | 59.7% | 84.0% | 33.0% |
| 8 | 87.8% | 68.3% | 99.6% | 49.7% | 98.0% | 62.5% | 83.8% | 36.4% |
| 9 | 80.6% | 73.9% | 99.6% | 46.3% | 97.4% | 60.2% | 83.9% | 36.5% |
| 10 | 74.6% | 74.9% | 99.5% | 45.7% | 97.1% | 61.8% | 83.8% | 40.4% |

Table 6: HMC samples for ReLU networks are most accurate for smaller numbers of layers, the samples from deeper models may therefore be slightly less reliable. Acceptance rates tend to be with 10-20 percentage points of 65%, regarded as a good balance of exploration to avoiding unnecessary resampling. The more linear models are less accurate, as one would expect for a dataset that is not linearly separable.

Figure 8: Example density for randomly chosen parameter from a ReLU network with three hidden layers. The HMC histogram is multimodal. If we picked the naive Gaussian fit, we would lie between the modes. By using a mixture model, we select the dominant mode, for which the Gaussian is a better fit.

Figure 9: CIFAR-100. Accuracy for diagonal and low-rank covariance SWAG. Like log-likelihood, there is no clear difference in performance between these models, all of which are above the depth threshold implied by our work.

We use the simple two-dimensional binary classification 'make moons' task.[4] We use 500 training points (generated using random_state = 0). Using Cobb et al. [2019], we apply the No-U-turn Sampling scheme [Hoffman and Gelman, 2014] with an initial step size of 0.01. We use a burn-in phase with 10,000 steps targeting a rejection rate of 0.8. We then sample until we collect 1,000 samples from the true posterior, taking 100 leapfrog steps in between every sample used in order to ensure samples are less correlated. For each result, we recalculate the HMC samples 20 times with a different random seed. All chains are initialized at the optimum of a mean-field variational inference model in order to help HMC rapidly find a mode of the true posterior. We use a prior precision, normalizing constant, and $\tau$ of 1.0. The model is designed to have as close to 1000 non-bias parameters each time as possible, adjusting the width given the depth of the model. We observe that the accuracies for the ReLU network fall for the deeper models, suggesting that after about 7 layers the posterior estimate may become slightly less reliable (see Table 6). Acceptance rates are broadly in a sensible region for most of the chains.

Using these samples, we find a Gaussian fit. For each model we fit a Gaussian mixture model with between 1 and 4 components and pick the one with the best Bayesian information criterion (see Figure 8). We then find the best diagonal fit to this distribution, which is a Gaussian distribution with the same mean and with a precision matrix equal to the inverse of the diagonal precision of the full-covariance Gaussian. We do this because variational inference uses the mode-seeking KL-divergence, so we are interested in the properties of a single Gaussian mode. The overall empirical covariance would lead to a mode-covering distribution, while optimizing the mode-seeking KL to the empirical distribution would of course result in a point-sized distribution centred at one of the HMC samples. Using one mode of a mixture of Gaussians is therefore the closest we can come to finding a single mode of the true posterior of the sort that VI might uncover. Note that we are therefore only considering one of the many modes of the true posterior—this is inevitable given the fact that there is a many-to-one correspondence between weight-distributions and function-distributions for neural networks.

Finally, we calculate the KL divergence between these two distributions. The graph reports the mean and shading reflects the standard error of the mean, though note that because all runs are initialized from the same point, this underestimates the overall standard error. All experiments in this an other sections were run on a desktop workstation with an Nvida 1080 GPU.

For the Wasserstein divergence, we estimate the distance between the empirical distributions formed by the HMC samples and samples from the full- and diagonal-covariance posterior approximation. We used the Python Optimal Transport package to estimate the divergence.

### C.4 Diagonal- and Structured-SWAG at Varying Depths

We use the implementation of SWAG avaliable publicly at `https://github.com/wjmaddox/swa_gaussian`. We adapt their code to vary the depth of the PreResNet architecture for the values $2, 8, 14, 20, 26, 32, 38$. We use the hyperparameter settings used by the Maddox et al. [2019] for PreResNet154 on CIFAR100 on all datasets to train the models. We use 10 seeds to generate the error bars, which are plotted with one standard deviation. We use the same SWAG run to fit both the full and diagonal approximations, and use 30 samples in the forward pass.

### C.5 Large Scale Experiments Descriptions

In Table 7 we show a complete version of Table 2 including the standard deviations over three runs (except for the Noisy K-FAC result where standard deviation was not provided). The standard deviations, of course, underestimate the true variability of the method in question on Imagenet as they only consider difference in random seed with the training configuration otherwise identical. Fuller descriptions of the experimental settings used by the authors are provided in the cited papers.

For CIFAR-10, we show similar results in Table 8. Here, authors compare a wider range of architectures, which show substantial variation in resulting accuracy. However, within the same architecture, there is little evidence of systematic differences between mean-field and structured-covariance methods and any differences which do appear are marginal. Note that Zhang et al. [2018] report difficulty applying batch normalization to mean-field methods, but Osawa et al. [2019] report no difficulties applying batch normalization for their mean-field variant of Noisy Adam. For this reason, we report the version of Noisy KFAC run without batch normalization to make it comparable with the results shown for Bayes-by-Backprop (BBB) and Noisy Adam. With batch normalization, Noisy KFAC gains some accuracy, reaching 92.0%, but this seems to be because of the additional regularization, not a property of the approximate posterior family.

## D Proofs

### D.1 Full Derivation of the Product Matrix Covariance

**Proposition 1.** *For $L \geq 3$, the product matrix $M^{(L)}$ of factorized weight matrices can have non-zero covariance between any and all pairs of elements. That is, there exists a set of mean-field weight matrices $\{W^{(l)} | 1 \leq l < L\}$ such that $M^{(L)} = \prod W^{(l)}$ and the covariance between any possible pair of elements of the product matrix:*

$$\text{Cov}(m_{ab}^{(L)}, m_{cd}^{(L)}) \neq 0, \tag{1}$$

| Architecture | Method | Covariance | Accuracy | NLL | ECE |
|---|---|---|---|---|---|
| ResNet-18 | VOGN[‡] | Diagonal | 67.4% ± 0.263 | 1.37 ± 0.010 | 0.029 ± 0.001 |
| ResNet-18 | Noisy K-FAC[††] | MVG | 66.4% ± n.d. | 1.44 ± n.d. | 0.080 ± n.d. |
| DenseNet-161 | SWAG-Diag[†] | Diagonal | 78.6% ± 0.000 | 0.86 ± 0.000 | 0.046 ± 0.000 |
| DenseNet-161 | SWAG[†] | Low-rank + Diag | 78.6% ± 0.000 | 0.83 ± 0.000 | 0.020 ± 0.000 |
| ResNet-152 | SWAG-Diag[†] | Diagonal | 80.0% ± 0.000 | 0.86 ± 0.000 | 0.057 ± 0.000 |
| ResNet-152 | SWAG[†] | Low-rank + Diag | 79.1% ± 0.000 | 0.82 ± 0.000 | 0.028 ± 0.000 |

Table 7: Imagenet. Comparison of diagonal-covariance/mean-field (in grey) and structured-covariance methods on Imagenet. The differences on a given architecture between comparable methods is slight. [†] [Maddox et al., 2019]. [‡] [Osawa et al., 2019]. [††] [Zhang et al., 2018] as reported by Osawa et al. [2019].

| Architecture | Method | Covariance | Accuracy | NLL | ECE |
|---|---|---|---|---|---|
| VGG-16 | SWAG-Diag[†] | Diagonal | 93.7% ± 0.15 | 0.220 ± 0.008 | 0.027 ± 0.003 |
| VGG-16 | SWAG[†] | Low-rank + Diag | 93.6% ± 0.10 | 0.202 ± 0.003 | 0.016 ± 0.003 |
| VGG-16 | Noisy Adam[‡‡] | Diagonal | 88.2% ± n.d. | n.d. | n.d. |
| VGG-16 | BBB[‡‡] | Diagonal | 88.3% ± n.d. | n.d. | n.d. |
| VGG-16 | Noisy KFAC[‡‡] | MVG | 89.4% ± n.d. | n.d. | n.d. |
| PreResNet-164 | SWAG-Diag[†] | Diagonal | 96.0% ± 0.10 | 0.125 ± 0.003 | 0.008 ± 0.001 |
| PreResNet-164 | SWAG[†] | Low-rank + Diag | 96.0% ± 0.02 | 0.123 ± 0.002 | 0.005 ± 0.000 |
| WideResNet28x10 | SWAG-Diag[†] | Diagonal | 96.4% ± 0.08 | 0.108 ± 0.001 | 0.005 ± 0.001 |
| WideResNet28x10 | SWAG[†] | Low-rank + Diag | 96.3% ± 0.08 | 0.112 ± 0.001 | 0.009 ± 0.001 |
| ResNet-18 | VOGN[‡] | Diagonal | 84.3% ± 0.20 | 0.477 ± 0.006 | 0.040 ± 0.002 |
| AlexNet | VOGN[‡] | Diagonal | 75.5% ± 0.48 | 0.703 ± 0.006 | 0.016 ± 0.001 |

Table 8: CIFAR-10. For a given architecture, it does not seem that mean-field (grey) methods systematically perform worse than methods with structured covariance, although there is some difference in the results reported by different authors. [†] [Maddox et al., 2019]. [‡] [Osawa et al., 2019]. [‡‡] [Zhang et al., 2018].

where $m_{ij}^{(L)}$ are elements of the product matrix in the $i^{th}$ row and $j^{th}$ column, and for any possible indexes $a$, $b$, $c$, and $d$.

*Proof.* We begin by explicitly deriving the covariance between elements of the product matrix.

Consider the product matrix, $M^{(L)}$, which is the matrix product of an arbitrary weight matrix, $W^{(L)}$, with a mean field distribution over it's entries, and the product matrix with one fewer layers, $M^{(L-1)}$. Expressed in terms of the elements of each matrix in row-column notation this matrix multiplication can be written:

$$m_{ab}^{(L)} = \sum_{i=1}^{K_{L-1}} w_{ai}^{(L)} m_{ib}^{(L-1)}. \tag{8}$$

We make no assumption about $K_{L-1}$ except that it is non-zero and hence the weights can be any rectangular matrix.[5] The weight matrix $W^{(L)}$ is assumed to have a mean-field distribution (the covariance matrix is zero for all off diagonal elements) with arbitrary means:

$$\mathrm{Cov}\big(w_{ac}^{(L)}, w_{bd}^{(L)}\big) = \Sigma_{abcd}^{(L)} = \delta_{ac}\delta_{bd}\sigma_{ab}^{(L)};$$
$$\mathbb{E}\, w^{(L)}{}_{ab} = \mu_{ab}^{(L)}. \tag{9}$$

$\delta$ are the Kronecker delta. Note that the weight matrix is 2-dimensional, but the covariance matrix is defined between every element of the weight matrix. While it can be helpful to regard it as 2-dimensional also, we index it with the four indices that define a pair of elements of the weight matrix.

We begin by deriving the expression for the covariance of the $L$-layer product matrix $\mathrm{Cov}(m_{ab}^{(L)}, m_{cd}^{(L)})$. Using the definition of the product matrix in equation (8):

$$\mathrm{Cov}\big(m_{ab}^{(L)}, m_{cd}^{(L)}\big) = \mathrm{Cov}\Big(\sum_i w_{ai}^{(L)} m_{ib}^{(L-1)}, \sum_j w_{cj}^{(L)} m_{jd}^{(L-1)}\Big). \tag{10}$$

We then simplify this using the linearity of covariance (for brevity call the covariance of the product matrix $\hat{\Sigma}_{abcd}^{(L)}$):

$$\hat{\Sigma}_{abcd}^{(L)} = \sum_{ij} \mathrm{Cov}\big(w_{ai}^{(L)} m_{ib}^{(L-1)}, w_{cj}^{(L)} m_{jd}^{(L-1)}\big), \tag{11}$$

rewriting using the definition of covariance in terms of a difference of expectations:

$$= \sum_{ij} \mathbb{E}\big[w_{ai}^{(L)} m_{ib}^{(L-1)} w_{cj}^{(L)} m_{jd}^{(L-1)}\big] - \mathbb{E}\big[w_{ai}^{(L)} m_{ib}^{(L-1)}\big] \mathbb{E}\big[w_{cj}^{(L)} m_{jd}^{(L-1)}\big], \tag{12}$$

using the fact that by assumption the new layer is independent of the previous product matrix:

$$= \sum_{ij} \mathbb{E}\big[w_{ai}^{(L)} w_{cj}^{(L)}\big] \mathbb{E}\big[m_{ib}^{(L-1)} m_{jd}^{(L-1)}\big] - \mathbb{E}\big[w_{ai}^{(L)}\big] \mathbb{E}\big[w_{cj}^{(L)}\big] \mathbb{E}\big[m_{ib}^{(L-1)}\big] \mathbb{E}\big[m_{jd}^{(L-1)}\big], \tag{13}$$

and rewriting to expose the dependence on the covariance of $M^{(L-1)}$:

$$\begin{aligned}
= \sum_{ij} & \Big( \mathbb{E}\big[w_{ai}^{(L)} w_{cj}^{(L)}\big] - \mathbb{E}\big[w_{ai}^{(L)}\big] \mathbb{E}\big[w_{cj}^{(L)}\big] \Big) \\
& \cdot \Big( \mathbb{E}\big[m_{ib}^{(L-1)} m_{jd}^{(L-1)}\big] - \mathbb{E}\big[m_{ib}^{(L-1)}\big] \mathbb{E}\big[m_{jd}^{(L-1)}\big] \Big) \\
& + \mathbb{E}\big[w_{ai}^{(L)}\big] \mathbb{E}\big[w_{cj}^{(L)}\big] \Big( \mathbb{E}\big[m_{ib}^{(L-1)} m_{jd}^{(L-1)}\big] - \mathbb{E}\big[m_{ib}^{(L-1)}\big] \mathbb{E}\big[m_{jd}^{(L-1)}\big] \Big) \\
& + \mathbb{E}\big[m_{ib}^{(L-1)}\big] \mathbb{E}\big[m_{jd}^{(L-1)}\big] \Big( \mathbb{E}\big[w_{ai}^{(L)} w_{cj}^{(L)}\big] - \mathbb{E}\big[w_{ai}^{(L)}\big] \mathbb{E}\big[w_{cj}^{(L)}\big] \Big),
\end{aligned} \tag{14}$$

substituting the covariance:

$$\begin{aligned}
= \sum_{ij} & \mathrm{Cov}\Big(w_{ai}^{(L)}, w_{cj}^{(L)}\Big) \cdot \mathrm{Cov}\Big(m_{ib}^{(L-1)}, m_{jd}^{(L-1)}\Big) \\
& + \mathbb{E}\big[w_{ai}^{(L)}\big] \mathbb{E}\big[w_{cj}^{(L)}\big] \mathrm{Cov}\Big(m_{ib}^{(L-1)}, m_{jd}^{(L-1)}\Big) \\
& + \mathbb{E}\big[m_{ib}^{(L-1)}\big] \mathbb{E}\big[m_{jd}^{(L-1)}\big] \mathrm{Cov}\Big(w_{ai}^{(L)}, w_{cj}^{(L)}\Big).
\end{aligned} \tag{15}$$

This gives us a recursive expression for the covariance of the product matrix.

It is straightforward to substitute in our expressions for mean and variance in a mean-field network provided in equation (9), where we use the fact that the initial $M^{(1)}$ product matrix is just a single mean-field layer.

In this way, we show that:

$$\begin{aligned}
\hat{\Sigma}_{abcd}^{(2)} = \sum_{ij} & \Big( \delta_{ac} \delta_{ij} \sigma_{ai}^{(2)} \Big) \cdot \Big( \delta_{ij} \delta_{bd} \sigma_{ib}^{(1)} \Big) \\
& + \mu_{ai}^{(2)} \mu_{cj}^{(2)} \Big( \delta_{ij} \delta_{bd} \sigma_{ib}^{(1)} \Big) + \mathbb{E}\big[m_{ib}^{(1)}\big] \mathbb{E}\big[m_{jd}^{(1)}\big] \Big( \delta_{ac} \delta_{ij} \sigma_{ai}^{(2)} \Big)
\end{aligned} \tag{16}$$

$$= \sum_i \delta_{ac} \delta_{bd} \sigma_{ai}^{(2)} \sigma_{ib}^{(1)} + \delta_{bd} \mu_{ai}^{(2)} \mu_{ci}^{(2)} \sigma_{ib}^{(1)} + \delta_{ac} \mu_{ib}^{(1)} \mu_{id}^{(1)} \sigma_{ai}^{(2)}. \tag{17}$$

The first term of equation (17) has the Kronecker deltas $\delta_{ac} \delta_{bd}$ meaning that it contains diagonal entries in the covariance matrix. The second term has only $\delta_{bd}$ meaning it contains entries for the covariance between weights that share a column. The third term has only $\delta_{ac}$ meaning it contains entries for the covariance between weights that share a row.

This covariance of the product matrix already has some off-diagonal terms, but it does not yet contain non-zero covariance for weights that share neither a row nor a column.

But we can repeat the process and find $\hat{\Sigma}_{abcd}^{(3)}$ using equation (15) and our expression for $\hat{\Sigma}_{ibjd}^{(2)}$:

$$\hat{\Sigma}_{abcd}^{(3)} = \sum_{ij} \left( \delta_{ac}\delta_{ij}\sigma_{ai}^{(3)} \right) \cdot \hat{\Sigma}_{ibjd}^{(2)} + \mu_{ai}^{(3)}\mu_{cj}^{(3)}\hat{\Sigma}_{ibjd}^{(2)} + \mathbb{E}\left[m_{ib}^{(2)}\right]\mathbb{E}\left[m_{jd}^{(2)}\right]\left( \delta_{ac}\delta_{ij}\sigma_{ai}^{(3)} \right) \quad (18)$$

$$= \sum_{ij} \left( \delta_{ac}\delta_{ij}\sigma_{ai}^{(3)} \right) \cdot \sum_{k} \left( \delta_{ij}\delta_{bd}\sigma_{ik}^{(2)}\sigma_{kb}^{(1)} + \delta_{bd}\mu_{ik}^{(2)}\mu_{jk}^{(2)}\sigma_{kb}^{(1)} + \delta_{ij}\mu_{kb}^{(1)}\mu_{kd}^{(1)}\sigma_{ik}^{(2)} \right)$$

$$+ \mu_{ai}^{(3)}\mu_{cj}^{(3)} \cdot \sum_{k} \left( \delta_{ij}\delta_{bd}\sigma_{ik}^{(2)}\sigma_{kb}^{(1)} + \delta_{bd}\mu_{ik}^{(2)}\mu_{jk}^{(2)}\sigma_{kb}^{(1)} + \textcolor{red}{\delta_{ij}\mu_{kb}^{(1)}\mu_{kd}^{(1)}\sigma_{ik}^{(2)}} \right)$$

$$+ \mu_{ib}^{(2)}\mu_{jd}^{(2)}\left( \delta_{ac}\delta_{ij}\sigma_{ai}^{(3)} \right). \quad (19)$$

It is the term in red which has no factors of Kronecker deltas in any of the indices a, b, c, or d. It is therefore present in all elements of the covariance matrix of the product matrix, regardless of whether they share one or both index. This shows that, so long as the distributional parameters themselves are non-zero, the product matrix can have a fully non-zero covariance matrix for $L = 3$.

We note that there are many weight matrices for which the resulting covariance is non-zero everywhere—we think this is actually typical. Indeed, empirically, we found that for any network we cared to construct, we were unable to find covariances that *were* zero anywhere. However, for our existance proof, we simply note that for any matrix in which all the means are positive each term of the resulting expression is positive (the standard deviation parameters may be taken as positive without loss of generality). In that case, it is impossible that any term cancels with any other, so the resulting covariance is positive everywhere.

Last, we examine the recurrence relationship in equation 15. Once $\text{Cov}(m_{ib}^{(L-1)}, m_{jd}^{(L-1)}) \neq 0$ for all possible indices, the covariances between elements of $M^{(L)}$ may also be non-zero. Observe simply that if the means of the top weight matrix are positive, then each of the terms in equation 15 are positive, so it is impossible for any term to cancel out with any other. The fact that the elements of $M^{(L-1)}$ have non-zero covariances everywhere therefore entails that there is a weight matrix $W^{(L)}$ such that $M^{(L)}$ has non-zero covariance between all of its elements also, as required.

**Remark 3.** *Here, we show only an existance proof, and therefore we restrict ourselves to positive means and standard deviations to simplify the proof. In fact, we believe that non-zero covariance is the norm, rather than a special case, and found this in all our numerical simulations for both trained and randomly sampled models. However, we do not believe that (in the linear case) any covariance matrix can be created from a deep mean-field product.*

□

## D.2 Matrix Variate Gaussian as a Special Case of Three-Layer Product Matrix

We can gain insight into the richness of the possible covariances by considering the limited case of the product matrix $M^{(3)} = ABC$ where $B$ is a matrix whose elements are independent Gaussian random variables and $A$ and $C$ are deterministic. We note that this is a highly constrained setting, and that the covariances which can be induced with $A$ and $C$ as random variables have the more complex form shown in D.1. We can show the following:

**Proposition 2.** *The Matrix Variate Gaussian (Kronecker-factored) distribution is a special case of the distribution over elements of the product matrix. In particular, for $M^{(3)} = ABC$, $M^{(3)}$ is distributed as an MVG random variable when $A$ and $C$ are deterministic and $B$ has its elements distributed as fully factorized Gaussians with unit variance.*

*Proof.* Consider the product matrix $M^{(3)} = ABC$. where $B$ is a matrix whose elements are independent Gaussian random variables and $A$ and $C$ are deterministic. The elements of $B$ are distributed with mean $\mu_B$ and have a diagonal covariance matrix $\Sigma_B$.

We begin by recalling the property of the Kronecker product that:

$$\text{vec}(ABC) = (C^\top \otimes A)\text{vec}(B). \tag{20}$$

By definition $\text{vec}(M^{(3)}) = \text{vec}(ABC) = (C^\top \otimes A)\text{vec}(B)$. Because $C^\top \otimes A$ is deterministic, it follows from a basic property of the covariance that the covariance of the product matrix $\Sigma_{M^{(3)}}$ is given by:

$$\Sigma_{M^{(3)}} = (C^\top \otimes A)\Sigma_B(C^\top \otimes A)^\top. \tag{21}$$

Using the fact that the transpose is distributive over the Kronecker product, this is equivalent to:

$$\Sigma_{M^{(3)}} = (C^\top \otimes A)\Sigma_B(C \otimes A^\top). \tag{22}$$

Because we only want to establish that the family of distributions expressible contains the matrix variate Gaussians, we do not need to use all the possible freedom, and we can set $\Sigma_B = I$. In this special case:

$$\Sigma_{M^{(3)}} = (C^\top \otimes A)(C \otimes A^\top). \tag{23}$$

Using the mixed-product property, this is equivalent to:

$$\Sigma_{M^{(3)}} = (C^\top C) \otimes (AA^\top). \tag{24}$$

Now, we note that any positive semi-definite matrix can be written in the form $A = M^\top M$, so this implies that, defining the positive semi-definite matrices $V = C^\top C$ and $U = AA^\top$, we have that the covariance $\Sigma_{M^{(3)}}$ is of the form,

$$\Sigma_{M^{(3)}} = V \otimes U. \tag{25}$$

Similarly, we can consider the mean of the product matrix $\mu_{M^{(3)}}$. From equation 20, we can see that:

$$\mu_{M^{(3)}} = (C^\top \otimes A)\text{vec}(\mu_B). \tag{26}$$

But since we have not yet constrained $\mu_B$, it is clear that this allows us to set any $\mu_{M^{(3)}}$ we desire by choosing $\mu_B = (C^\top \otimes A)^{-1}\mu_{M^{(3)}}$.

So far, we have only discussed the first- and second-moments, and the proof has made no assumptions about specific distributions. However, we now observe that a random variable $X$ is distributed according to the Matrix Variate Gaussian distribution according to some mean $\mu_X$ and with scale matrices $U$ and $V$ if and only if $\text{vec}(X)$ is a multivariate Gaussian with mean $\vec{\mu_X}$ and covariance $U \otimes V$.

Therefore, given equations (26) and (25), the special case of $M^{(3)}$ where the first and last matrices are deterministic and the middle layer has a fully-factorized Gaussian distribution over the weights with unit variance is a Matrix Variate Gaussian distribution where:

$$\text{vec}(\mu_X) = (C^\top \otimes A)\text{vec}(\mu_B); \tag{27}$$

$$V = C^\top C; \tag{28}$$

$$U = A^\top A. \tag{29}$$

$\square$

## D.3  Distribution of the Product Matrix

In general the probability density function of a product of random variables is not the product of their density functions. In the scalar case, the product of two independent Gaussian distributions is a generalized $\chi^2$ distribution. The product of arbitrarily many Gaussians with arbitrary non-i.i.d. mean and variance is difficult to calculate (special cases are much better understood e.g., Springer and Thompson [1970]). An example of a distribution family that *is* closed under multiplication is the log-normal distribution.

In the case of matrix multiplication, important for neural network weights, because each element of a product of matrix multiplication is the sum of the product of individual elements we would ideally like a distribution to be closed under both addition and multiplication (such as the Generalized Gamma convolution [Bondesson, 2015]) but these are not practical.

Instead, it would be helpful if we could make use of a simple distribution like the Gaussian but maintain roughly similar distributions over product matrix elements as the network becomes deeper. For only one layer of hidden units, provided K is sufficiently large, we can use the central limit theorem to show that the elements of the product matrix composed of i.i.d. Gaussian priors tends to a Gaussian as the width of the hidden layer increases. For two or more layers, however, the central limit theorem fails because the elements of the product matrix are no longer independent. However, even though the resulting product matrix is not a Gaussian, we show through numerical simulation that products of matrices with individual weights distributed as $\mathcal{N}(0, 0.23^2)$ have roughly the same distribution over their weights. This, combined with the fact that our choice of Gaussian distributions over weights was somewhat arbitrary in the first place, might reassure us that the increase in depth does not change the model prior in an important way. In Figure 10 we plot the probability density function of an arbitrarily chosen entry in the product matrix with varying depths of diagonal Gaussian prior weights. The p.d.f. for 7 layers is approximately the same as the single-layer Gaussian distribution with variance $0.23^2$.

Figure 10: Density over arbitrary element of product matrix for $L$ diagonal prior Gaussian weight matrices whose elements are i.i.d. $\mathcal{N}(0, 0.23^2)$. Product matrix elements are not strictly Gaussian, but very close.

## D.4 Proof of Linearized Product Matrix Covariance

### D.4.1 Proof of Local Linearity

We consider local linearity in the case of piecewise-linear activations like ReLU.

**Lemma 1.** *Consider an input point $\mathbf{x}^* \in \mathcal{D}$. Consider a realization of the model weights $\boldsymbol{\theta}$. Then, for any $\mathbf{x}^*$, the neural network function $f_{\boldsymbol{\theta}}$ is linear over some compact set $\mathcal{A}_{\boldsymbol{\theta}} \subset \mathcal{D}$ containing $\mathbf{x}^*$. Moreover, $\mathcal{A}_{\boldsymbol{\theta}}$ has non-zero measure for almost all $\mathbf{x}^*$ w.r.t. the Lebesgue measure.*

*Proof.* Neural networks with finitely many piecewise-linear activations are themselves piecewise-linear. Therefore, for a finite neural network, we can decompose the input domain $\mathcal{D}$ into regions $\mathcal{D}_i \subseteq \mathcal{D}$ such that

1. $\cup \mathcal{D}_i = \mathcal{D}$,

2. $\mathcal{D}_i \cap \mathcal{D}_j = \varnothing \quad \forall i \neq j$,

3. $f_{\boldsymbol{\theta}}$ is a linear function on points in $\mathcal{D}_i$ for each i.

For a finite neural network, there are at most finitely many regions $\mathcal{D}_i$. In particular, with hidden layer widths $n_i$ in the $i$'th layer, with an input domain $\mathcal{D}$ with dimension $n_0$, Montúfar et al. [2014] show that the network can define maximally a number of regions in input space bounded above by:

$$\left( \prod_{i=1}^{L-1} \left\lfloor \frac{n_i}{n_0} \right\rfloor^{n_0} \right) \sum_{j=0}^{n_0} \binom{n_L}{j}. \tag{30}$$

Except in the trivial case where the input domain has measure zero, this along with (1) and (2) jointly entail that at least one of the regions $\mathcal{D}_i$ has non-zero measure. This, with (3) entails that only a set of input points of zero measure do *not* fall in a linear region of non-zero measure. These points correspond to inputs that lie directly on the inflection points of the ReLU activations. $\square$

(a) 1 sample: $\mathcal{A}_{\boldsymbol{\theta}_i}$

(b) 5 samples: $\mathcal{A} = \bigcap_{0 \le i < 5} \mathcal{A}_{\boldsymbol{\theta}_i}$

Figure 11: Visualizaton of the linear regions in input-space for a two-dimensional binary classification problem (two moons). Colored regions show contiguous areas within which a neural network function is linear. We use an abitrary numerical encoding of these regions (we interpret the sign pattern of activated relus as an integer in base 2) and a cylic colour scheme for visualisation, so the color of each region is arbitrary, and two non-contiguous regions with the same color are not the same region. The neural network has one hidden layer with 100 units and is trained for 1000 epochs on 500 datapoints from scipy's two moons using Adam. (a) a single model has fairly large linear regions, with the most detail clustered near the region of interest. (b) The regions within which all samples are linear (the intersection set $\mathcal{A}$) are smaller, but finite. The local product matrix is valid within one of these regions for any input point.

We visualize $\mathcal{A}_{\boldsymbol{\theta}_i}$ in Figure 11a. This shows a two-dimensional input space (from the two moons dataset). Parts of the space within which a neural network function is linear are shown in one color. The regions are typically smallest where the most detail is required in the trained function.

### D.4.2 Defining the Local Product Matrix

We define a random variate representing the local product matrix, for an input point $\mathbf{x}^*$, using the following procedure.

To draw a finite $N$ samples of the random variate, we sample $N$ realizations of the weight parameters $\Theta = \{\boldsymbol{\theta}_i \text{ for } 1 \le i \le N\}$. For each $\boldsymbol{\theta}_i$, given $\mathbf{x}^*$ there is a compact set $\mathcal{A}_{\boldsymbol{\theta}_i} \subset \mathcal{D}$ within which $f_{\boldsymbol{\theta}_i}$ is linear (and $\mathbf{x}^* \in \mathcal{A}_i$) by lemma 1. Therefore, all samples of the neural network function are linear in the intersection region $\mathcal{A} = \bigcap_i \mathcal{A}_{\boldsymbol{\theta}_i}$. We note that $\mathcal{A}$ at least contains $\mathbf{x}^*$. Moreover, so long as $\mathcal{D}$ is a compact subset of the reals, $\mathcal{A}$ has non-zero measure.[6] Figure 11b shows a visualization of $\mathcal{A}$ with 5 samples. The linear regions are smaller, because there is a discontinuity if any of the models is

More formally, consider some compact set $\mathcal{A}_{\boldsymbol{\theta}_0} \subset \mathcal{D}$ with non-zero measure such that $\mathbf{x}^* \in \mathcal{A}_{\boldsymbol{\theta}_0}$. Take some new compact set $\mathcal{A}_{\boldsymbol{\theta}_1} \subset \mathcal{D}$ with non-zero measure also such that $\mathbf{x}^* \in \mathcal{A}_{\boldsymbol{\theta}_1}$. Define the intersection between those sets $\mathcal{B} = \mathcal{A}_{\boldsymbol{\theta}_0} \cap \mathcal{A}_{\boldsymbol{\theta}_1}$. Suppose that $\mathcal{B}$ has zero measure. But both $\mathcal{A}_{\boldsymbol{\theta}_0}$ and $\mathcal{A}_{\boldsymbol{\theta}_1}$ contain $\mathbf{x}^*$, so the only way that $\mathcal{B}$ could have zero measure is if $\mathbf{x}^*$ is an element in the boundary of both sets. But if $\mathcal{A}_{\boldsymbol{\theta}_1}$ has $\mathbf{x}^*$ on its boundary, then, by the continuity of the real space, there is at least one other compact set $\mathcal{A}'_{\boldsymbol{\theta}_1}$, different to $\mathcal{A}_{\boldsymbol{\theta}_1}$, such that $\mathbf{x}^*$ is on its boundary. But, since by hypothesis $\mathcal{A}_{\boldsymbol{\theta}_0}$ has non-zero measure, there exists such a set $\mathcal{A}'_{\boldsymbol{\theta}_1}$ which has a non-zero-measure intersection with $\mathcal{A}_{\boldsymbol{\theta}_0}$. We can therefore select $\mathcal{A}'_{\boldsymbol{\theta}_1}$ instead of $\mathcal{A}_{\boldsymbol{\theta}_1}$ when building $\mathcal{A}$, such that the intersection with $\mathcal{A}_{\boldsymbol{\theta}_0}$ has non-zero measure. By repeated application of this argument, we can guarantee that for any finite $\Theta$ we are able to find a set of $\mathcal{A}_{\boldsymbol{\theta}_i} \subset \mathcal{D}$ such that $\forall i : \mathbf{x}^* \in \mathcal{A}_{\boldsymbol{\theta}_i}$ and $\mathcal{A}$ has non-zero measure. This argument does not guarantee that the measure of $\mathcal{A}$ in the limit as $N$ tends to infinity is non-zero.

discontinuous. Nevertheless, the space is composed of regions of finite size within which the neural network function is linear.

For each $\boldsymbol{\theta}_i$ we can compute a local product matrix within $\mathcal{A}$. Ordinarily, setting aside the bias term for simplicity, a neural network hidden layer $\mathbf{h}_{l+1}$ can be written in terms of the hidden layer before it, a weight matrix $W_l$, and an activation function.

$$\mathbf{h}_{l+1} = \sigma(W_l \mathbf{h}_l) \tag{31}$$

We observe that within $\mathcal{A}$ the activation function becomes linear. This allows us to define an activation vector $\mathbf{a}_{\mathbf{x}^*}$ within $\mathcal{A}$ such that the equation can be written:

$$\mathbf{h}_{l+1} = \mathbf{a}_{\mathbf{x}^*} \cdot (W_l \mathbf{h}_l). \tag{32}$$

The activation vector can be easily calculated by calculating $W_l \mathbf{h}_l$, seeing which side of the (Leaky) ReLU the activation is on within that linear region for each hidden unit, and selecting the correct scalar (0 or 1 for a ReLU, or $\alpha$ or 1 for a Leaky ReLU).

This allows us to straightforwardly construct a product matrix for each $\boldsymbol{\theta}_i$ which takes the activation function into account (in the linear case, we effectively always set $\mathbf{a}_{\mathbf{x}^*}$ to equal the unit vector). The random variate $P_{\mathbf{x}^*}$ is constructed with these product matrices for realizations of the weight distribution.

Samples from the resulting random variate $P_{\mathbf{x}^*}$ are therefore distributed such that samples from $P_{\mathbf{x}^*}\mathbf{x}^*$ have the same distribution as samples of the predictive posterior $y$ given $\mathbf{x}^*$ within $\mathcal{A}$.

### D.4.3 Proof that the Local Product Matrix has Non-zero Off-diagonal Covariance

**Proposition 3.** *Given a mean-field distribution over the weights of neural network $f$ with piecewise linear activations, $f$ can be written in terms of the local product matrix $P_{\mathbf{x}^*}$ within $\mathcal{A}$.*

*For $L \geq 3$, for activation functions which are non-zero everywhere, there exists a set of weight matrices $\{W^{(l)} | 1 \leq l < L\}$ such that all elements of the local product matrix have non-zero off-diagonal covariance:*

$$\mathrm{Cov}(p_{ab}^{\mathbf{x}^*}, p_{cd}^{\mathbf{x}^*}) \neq 0, \tag{2}$$

*where $p_{ij}^{\mathbf{x}^*}$ is the element at the $i^{th}$ row and $j^{th}$ column of $P_{\mathbf{x}^*}$.*

*Proof.* First, we show that the covariance between arbitrary entries of each realization of the product matrix of linearized functions can be non-zero. Afterwards, we will show that this implies that the covariance between arbitrary entries of the product matrix random variate, $P_{\mathbf{x}^*}$ can be non-zero.

Consider a local product matrix constructed as above. Then for each realization of the weight matrices, the product matrix realization $M_i^{(L)}$, defined in the region around $\mathbf{x}^*$ following lemma 1. We can derive the covariance between elements of this product matrix within that region in the same way as in Proposition 1, finding similarly that:

$$
\begin{aligned}
\hat{\Sigma}_{abcd}^{(3)} = \alpha_{abcd} \sum_{ij} & \left( \delta_{ac}\delta_{ij}\sigma_{ai}^{(3)} \right) \cdot \sum_{k} \left( \delta_{ij}\delta_{bd}\sigma_{ik}^{(2)}\sigma_{kb}^{(1)} \right. \\
& + \delta_{bd}\mu_{ik}^{(2)}\mu_{jk}^{(2)}\sigma_{kb}^{(1)} \\
& \left. + \delta_{ij}\mu_{kb}^{(1)}\mu_{kd}^{(1)}\sigma_{ik}^{(2)} \right) \\
+ \mu_{ai}^{(3)}\mu_{cj}^{(3)} \cdot \sum_{k} & \left( \delta_{ij}\delta_{bd}\sigma_{ik}^{(2)}\sigma_{kb}^{(1)} \right. \\
& + \delta_{bd}\mu_{ik}^{(2)}\mu_{jk}^{(2)}\sigma_{kb}^{(1)} \\
& \left. + \delta_{ij}\mu_{kb}^{(1)}\mu_{kd}^{(1)}\sigma_{ik}^{(2)} \right) \\
+ \mu_{ib}^{(2)}\mu_{jd}^{(2)} & \left( \delta_{ac}\delta_{ij}\sigma_{ai}^{(3)} \right),
\end{aligned} \tag{33}
$$

where $\alpha$ is a constant determined by the piecewise-linearity in the linear region we are considering. Note that we must assume here that $\alpha_{ij} \neq 0$ except in a region of zero measure, for example a LeakyReLU, otherwise it is possible that the constant introduced by the activation could eliminate non-zero covariances. We discuss this point further below.

Now note that the covariance of the sum of independent random variables is the sum of their covariances. Therefore the covariance of $I$ realizations of $P$ (suppressing the notation $(L)$) is:

$$\text{Cov}(P_{ab}, P_{cd}) = \frac{1}{I} \sum_{i=1}^{I} \hat{\Sigma}_{abcd}^i \tag{34}$$

As before, consider the case of positive means and standard deviations. Just like before, this results in a positive entry in the covariance between any two elements for each realization of the product matrix, and by equation 34 the entry for any element of the local product matrix remains positive as well. This suffices to prove the proposition for the case of $L = 3$. Just as Proposition 1 extends to all larger $L$, this result does also.

**Remark 4.** *The above proof assumes that $\alpha_{ij} \neq 0$. In fact, for common piecewise non-linearities like ReLU, $\alpha$ may indeed be zero. This means that the non-linearity can, in principle, 'disconnect' regions of the network such that the 'effective depth' falls below 3 and there is not a covariance between every element.*

*We cannot rule this out theoretically, as it depends on the data and learned function. In practice, we find it very unlikely that a trained neural network will turn off all its activations for any typical input, nor that enough activations will be zero that the product matrix does not have shared elements after some depth.*

*However, we do observe that for at least some network structures and datasets it is uncommon in practice that all the activations in several layers are 'switched off'. We show in Figure 1 an example of a local product matrix covariance which does not suffer from this problem. We find that for a model trained with mean-field VI on the FashionMNIST test dataset the number of activations switched on is on average 48.5% with standard deviation 4.7%. There were only four sampled models out of 100 samples on each of 10,000 test points where an entire row of activations was 'switched off', reducing the effective depth by one, and this never occurred in more than one row. Indeed, [Goldblum et al., 2019] describe settings with all activations switched off as a pathological case where SGD fails.*

$\square$

### D.5 Existence Proof of a Two-Hidden-Layer Mean-field Approximate Posterior Inducing the True Posterior Predictive

In this section we prove that:

**Proposition 4.** *Let $p(\mathbf{y} = \mathbf{Y}|\mathbf{x}, \mathcal{D})$ be the probability density function for the posterior predictive distribution of any given multivariate regression function, with $\mathbf{x} \in \mathcal{A}$ where $\mathcal{A}$ is a compact set in $\mathbb{R}^D$, $\mathbf{y} \in \mathbb{R}^K$, and $\mathbf{Y}$ the posterior predictive random variable. Let $f(\cdot)$ be a Bayesian neural network with two hidden layers. Let $\hat{\mathbf{Y}}$ be the random vector defined by $f(\mathbf{x})$. Then, for any $\epsilon, \delta > 0$, there exists a set of parameters defining the neural network $f$ such that the absolute value of the difference in probability densities for any point is bounded:*

$$\forall \mathbf{y}, \mathbf{x} \in \mathcal{A}, i: \quad \Pr\left(|p(y_i = \hat{Y}_i) - p(y_i = Y_i|\mathbf{x}, \mathcal{D})| > \epsilon\right) < \delta, \tag{3}$$

*so long as: the activations of $f$ are non-polynomial, non-periodic, and have only zero-measure non-monotonic regions, the first hidden layer has at least $D + 1$ units, the second hidden layer has an arbitrarily large number of units, the cumulative density function of the posterior predictive is continuous in output-space, and the probability density function is continous and finite non-zero everywhere. Here, the probability bound is with respect to the distribution over a subset of the weights described in the proof, $\boldsymbol{\theta}_{\text{Pr}}$, while one weight distribution $\boldsymbol{\theta}_Z$ induces the random variable $\hat{\mathbf{Y}}$.*

*Proof.* We extend an informal construction by Gal [2016] which aimed to show that a sufficiently deep network with a unimodal approximate posterior could induce a multi-modal posterior predictive

by learning the inverse cumulative distribution function (c.d.f.) of the multi-modal distribution. In our case, we are not chiefly interested in the number of modes, but more generally the expressive power of the mean-field distribution in a BNN of sufficient width and depth. First, we outline a simplified version of the proof that highlights the main mechanisms involved but is not constructed with a Bayesian neural network. Later, we prove the full result for Bayesian neural networks.

### D.5.1 Simplified Construction

Figure 12: The random variable whose probability density function is the true predictive posterior $p(y|\mathbf{x}, \mathcal{D})$ can be written $Y|\mathbf{x}$. If it has an inverse cumulative density function (c.d.f.), $F_{Y|\mathbf{x}}^{-1}$, we can transform a uniform random variable $U$ onto it. We can approximate this inverse c.d.f. with $f_{\mathbf{x}}$ indexed by $\mathbf{x}$. The random variable given by $\hat{Y} := f_{\mathbf{x}}(U)$ can be constructed to be 'similar' to $Y|\mathbf{x}$ as we show.

**Lemma 2.** *Let $p(y = Y|\mathbf{x}, \mathcal{D})$ be the probability density function for the posterior predictive distribution of any given univariate regression function. Let $U$ be a uniformly distributed random variable. Let $f(\cdot)$ be a deterministic neural network with a single hidden layer of arbitrary width and invertible non-polynomial activations. Let $\hat{Y}$ be the random variable defined by $f(U, \mathbf{x})$. Then, for any $\epsilon > 0$, there exists a set of parameters defining a sufficiently wide $f$ such that the absolute value of the difference in probability densities for any point is bounded:*

$$\forall y, \mathbf{x}: \quad |p(y = \hat{Y}) - p(y = Y|\mathbf{x}, \mathcal{D})| < \epsilon, \tag{35}$$

*so long as the cumulative distribution function of the posterior predictive is continuous in output-space and the probability density function is non-zero everywhere.*

*Proof.* An outline of the proof for the simplified case is shown in Figure 12.

Suppose there is a true posterior distribution over function outputs whose probability density function (p.d.f) is given by $p(y = Y|\mathbf{x}, \mathcal{D})$. These define a random variable that we will denote $Y|\mathbf{x}$.

We also have some approximation that takes $\mathbf{x}$ as an input and returns some $y$ in the output space. Later, this will be our Bayesian neural network, but for now we simplify. Instead, our procedure is to have some deterministic neural network which accepts as an input a realization of a uniformly distributed random variable, $U$, and the input point $\mathbf{x}$. We define a random variable $\hat{Y} := f(U, \mathbf{x})$.

We would like to show that it is possible to construct a neural network $f$ such that the result in equation (35) holds—that $\hat{Y}$ is suitably similar to the true predictive posterior random variable $Y|\mathbf{x}$.

First, note that if $Y|\mathbf{x}$ has an inverse cumulative density function (c.d.f.) then, by the universality of the uniform, transforming a uniform random variable by this function creates a random variable distributed as $Y|\mathbf{x}$. As a result, if there is such an invertible cumulative density function, there is also a function mapping $U$ and $\mathbf{x}$ onto $Y|\mathbf{x}$.

Second, consider the conditions under which $Y|\mathbf{x}$ has an invertible c.d.f. We must assume that the c.d.f. is continuous in output space and that the probability density function is non-zero everywhere. The first is reasonable for most normal problems, we often make a stronger assumption of Lipschitz continuity. The second is also relatively mild, corresponding to non-dogmatic certainty (a posterior distribution that puts zero probability density on some output given some input can never update away from that in light of new information). Given these mild assumptions, therefore, we know that there exists a continuous function $F_{Y|\mathbf{x}}^{-1}$ which is the inverse of the c.d.f. of $Y|\mathbf{x}$.

Third, we consider how we might approximate this function. Here, we invoke the universal approximation theorem (UAT) [Leshno et al., 1993]. This states that for any continuous function $g$, arbitrary

fixed error, $\epsilon$, and compact subset $\mathcal{A}$ of $\mathbb{R}^D$, there exists a deterministic neural network with an arbitrarily wide single layer of hidden units and a non-polynomial activation, $f$ such that:

$$\forall \mathbf{a} \in \mathcal{A} : |f(\mathbf{a}) - g(\mathbf{a})| < \epsilon. \tag{36}$$

By setting the arbitrary continuous function as the inverse c.d.f. of $Y|\mathbf{x}$, that is, $g(U, \mathbf{x}) = F_{Y|\mathbf{x}}^{-1}(U)$ (which we have already assumed is continuous) it follows that:

$$\forall u, \mathbf{x} \in \mathcal{A} : |f(u, \mathbf{x}) - F_{Y|\mathbf{x}}^{-1}(u)| < \epsilon, \quad u \sim U. \tag{37}$$

Fourth, we convert this bound on the inverse c.d.f. into a bound on the c.d.f. and then a bound on the p.d.f. We rewrite the function represented by the neural network to make explicit that we are using $\mathbf{x}$ to index a function from $u$ to $y$: $y = f(u, \mathbf{x}) = f_{\mathbf{x}}(u)$.

For this step, we will need to be able to invert the approximation to the inverse CDF such that $u = f_{\mathbf{x}}^{-1}(y)$ (for fixed $\mathbf{x}$). In general for neural network functions this is not true. As a result, we employ a construction which breaks apart the line over which $y$ runs into subsegments within which the network is invertible. For non-periodic activation functions which have only zero-measure non-monotonic regions (e.g., ReLU) there will be finitely many of these segments given a finite number of hidden units. Let us index over these subregions with $i$, noting that we can think of the distribution over $y$ as a weighted mixture distribution with a member for each subregion which has zero-density outside of that subregion. The approximate inverse CDF of each of these sub-region mixture members can be written as $f_{\mathbf{x}}^i(u)$ such that $f_{\mathbf{x}}(u) = \sum_i f_{\mathbf{x}}^i(u)$. Each of these $f_{\mathbf{x}}^i(u)$ is invertible. We can therefore rewrite equation (37), since $u$ and $\mathbf{x}$ are bounded by assumption, as:

$$\forall y, \mathbf{x} \in \mathcal{A} : \quad |\sum_i f_{\mathbf{x}}(f_{\mathbf{x}}^{i-1}(y)) - F_{Y|\mathbf{x}}^{-1}(f_{\mathbf{x}}^{i-1}(y))| < \epsilon. \tag{38}$$

which implies that:

$$\forall y, \mathbf{x} \in \mathcal{A} : \quad |\sum_i y - F_{Y|\mathbf{x}}^{-1}(f_{\mathbf{x}}^{i-1}(y))| < \epsilon. \tag{39}$$

Remember that we assumed above that the c.d.f. of $Y|\mathbf{x}$ is uniformly continuous, which means that for any $y'$ and $y''$, and any $\epsilon > 0$ there exists a $\delta$ such that if $|y' - y''| < \delta$ then $|F_{Y|\mathbf{x}}(y') - F_{Y|\mathbf{x}}(y'')| < \epsilon$. Alongside equation (39), and canceling the c.d.f. with the inverse c.d.f. this entails that:

$$\forall y, \mathbf{x} \in \mathcal{A} : \quad |\sum_i F_{Y|\mathbf{x}}(y) - f_{\mathbf{x}}^{i-1}(y))| < \epsilon. \tag{40}$$

But since $f_{\mathbf{x}}^{i-1}(y)$ is zero by construction outside of its subregion, this results in a bound on the overall c.d.f. of the random variable. That is to say, the bound on the inverse c.d.f. implies a bound on the c.d.f.

Finally, we remember that the cumulative density is the integral of the probability density function. Therefore, by Theorem 7.17 of Rudin [1976] and the uniform convergence in the c.d.f.s, it follows that:

$$\forall y, \mathbf{x} \in \mathcal{A} : \quad |p(y = Y|\mathbf{x}, \mathcal{D}) - p(y = \hat{Y})| < \epsilon \tag{41}$$

introducing a bound in the probability density functions of the random variable of true posterior outputs and the outputs of the approximation $\hat{Y} = f(U, \mathbf{x})$. $\qquad \square$

### D.5.2 Full Construction

The full construction extends the result above in the following ways:

- Rather than separately introducing $U$, we show how the first layer of a Bayesian neural network can map $\mathbf{x} \rightarrow \mathbf{x}', Z$, where $Z$ is a unit Gaussian random variable and $\mathbf{x}'$ is a noised version of $\mathbf{x}$.

- Rather than using a deterministic neural network for the universal approximation theorem, we apply the stochastic adaptation introduced by Foong et al. [2020].

- Rather than a univariate regression, we consider multivariate regression.

Like Leshno et al. [1993] we note that the extension from univariate to multivariate regression follows trivially from the existence of a mapping from $\mathbb{R} \to \mathbb{R}^K$.[7]

We first give some intuition as to how the proof works. The first weight layer serves to map $\mathbf{x} \to \mathbf{x}', Z$. This sets us up in a similar situation to the proof in the previous section, where we began with $\mathbf{x}, U$. This requires two small adjustments to the proof above. The first is that the random variable we introduce is now Gaussian, rather than Uniform. The second is that all our results will be in terms of $\mathbf{x}'$, rather than $\mathbf{x}$, and an additional step will be required to convert a probabilistic bound in one to the other (noting that we can freely set the weights in the first layer to have arbitrarily small variance).

The second weight layer will play the role of the neural network in the simplified proof. This also will require a small modification, because earlier we assumed that the neural network was deterministic, but it is now stochastic. This means that the final result becomes a probabilistic bound.

As a result of all of these changes, the proof becomes considerably more complicated, though nothing important changes in the intuition behind the construction.

**Step 1: Mapping $\mathbf{x} \to \mathbf{x}', Z$**

Consider inputs $\mathbf{x} \in \mathbb{R}^D$. We define a two-hidden-layer neural network with an invertible non-polynomial activation function $\phi : \mathbb{R} \mapsto \mathbb{R}$. The first component of the network is a single weight matrix mapping onto a vector of hidden units: $\mathbf{h}_1 = \phi(\boldsymbol{\theta}_0 \mathbf{x} + \mathbf{b}_0)$. The second component is a neural network with a layer of hidden units defined relative to the first layer of units $\mathbf{h}_2 = \phi(\boldsymbol{\theta}_1 \mathbf{h}_1 + \mathbf{b}_1)$, and outputs $\mathbf{y} = \boldsymbol{\theta}_2 \mathbf{h}_2 + \mathbf{b}_2$. The distribution over the outputs $\mathbf{y}$ defines the random variable $\hat{\mathbf{Y}}$. Here, $\boldsymbol{\theta}_0$, $\boldsymbol{\theta}_1$, and $\boldsymbol{\theta}_2$ are matrices of independent Gaussian random variables and $\mathbf{b}_0$, $\mathbf{b}_1$, and $\mathbf{b}_2$ are vectors of independent Gaussian random variables. Given some dataset $\mathcal{D}$ the predictive posterior distribution over outputs is $p(\mathbf{y}|\mathbf{x}, \mathcal{D})$ which we associate with the random variable $\mathbf{Y}$. This is our (intractable) target.

Consider only the first component of the neural network, which maps $\mathbf{x}$ onto $\mathbf{h}_1$. We can construct simple constraints on $\boldsymbol{\theta}_0$ and $\mathbf{b}_0$ such that:

$$\mathbf{h}_1 = \begin{pmatrix} \phi(z) \\ \phi(\mathbf{x}') \end{pmatrix}, \tag{42}$$

where $z \sim Z$, a unit Gaussian random variable, and $\mathbf{x}' \in \mathbb{R}^D$, such that $\Pr(\|\mathbf{x}' - \mathbf{x}\| > \epsilon_1) < \delta_1$. In particular, suppose that for $\boldsymbol{\theta}_0 \in \mathbb{R}^{D \times D+1}$ and $\mathbf{b}_0 \in \mathbb{R}^{D+1}$:

$$\boldsymbol{\theta}_0 = \mathcal{N}(M_{\boldsymbol{\theta}_0}, \Sigma_{\boldsymbol{\theta}_0}) \text{ where } M_{\boldsymbol{\theta}_0} = (\mathbf{0}_D, \mathbb{I}_{D \times D}); \Sigma_{\boldsymbol{\theta}_0} = \sigma^2 \mathbb{I}_{D \times D+1}; \tag{43}$$

$$\mathbf{b}_0 = \mathcal{N}(\boldsymbol{\mu}_{\mathbf{b}_0}, \boldsymbol{\sigma}_{\mathbf{b}_0}) \text{ where } \boldsymbol{\mu}_{\mathbf{b}_0} = \mathbf{0}_{D+1}, \boldsymbol{\sigma}_{\mathbf{b}_0} = \begin{pmatrix} 1 & \sigma & \dots & \sigma \end{pmatrix}. \tag{44}$$

By multiplication, straightforwardly $\mathbf{x}' = \mathcal{N}(\mathbf{x}, 2\sigma^2 \mathbf{1})$. It follows trivially that for any $\epsilon_1$ and $\delta_1$ there exists some $\sigma$ such that the bound holds. We will apply this bound at the end of the proof to convert a bound in $\mathbf{x}'$ to one in $\mathbf{x}$.

Here, we introduce a distinction between the weights which determine $\mathbf{x}'$ and those that create $Z$. The only weights which determine $Z$ are the first element of $\boldsymbol{\theta}_0$ and the first element of $\mathbf{b}_0$. Call these $\boldsymbol{\theta}_Z$. We then define the remainder of the weight distributions as $\boldsymbol{\theta}_{\mathrm{Pr}} \coloneqq \{\boldsymbol{\theta}_0, \boldsymbol{\theta}_1, \boldsymbol{\theta}_2, \mathbf{b}_0, \mathbf{b}_1, \mathbf{b}_2\} \setminus \boldsymbol{\theta}_Z$. This distinction is important, because the probabilistic bound in the proof will be over $\boldsymbol{\theta}_{\mathrm{Pr}}$ while the distribution over $\boldsymbol{\theta}_Z$ will induce the random variable $\hat{\mathbf{Y}}$.

**Step 2: Invoking a Result Similar to the Simplified Construction**

We show that there is a function which maps $Z$ and $\mathbf{x}'$ onto $\mathbf{Y}$, under reasonable assumptions similar to those of the simplified construction. For brevity, we denote the probability density function (p.d.f.) of the true posterior predictive distribtion of the random variable $\mathbf{Y}$ conditioned on $\mathbf{x}'$ and $\mathcal{D}$ as $f_{Y|\mathbf{x}'} \equiv p(Y = y|\mathbf{x}', \mathcal{D})$ and the cumulative density function (c.d.f.) as $F_{Y|\mathbf{x}'}$. We similarly write the inverse c.d.f. as $F_{Y|\mathbf{x}'}^{-1}$.

We must adapt the simplified construction to account for the fact that rather than simply approximating the inverse of the c.d.f. we now need to also transform the Gaussian random variable onto a Uniform one and invert the activation. We show below:

**Lemma 3.** *There exists continous function $G_{\mathbf{x}'}^{-1} = F_{Y|\mathbf{x}'}^{-1} \cdot F_Z \cdot \phi^{-1}$, where $F_Z$ is the c.d.f. of the unit Gaussian and $\phi^{-1}$ is the inverse of the activation function, such that the random variable $G_{\mathbf{x}'}^{-1}(\phi(Z))$ is equal in distribution to $\mathbf{Y}|\mathbf{x}$, if the p.d.f. of the posterior predictive is non-zero everywhere and the c.d.f. is continuous.*

The limitation to p.d.f.s is modest as before. The naming of the function $G_{\mathbf{x}'}^{-1}$ is suggestive, and indeed its inverse exists if $F_{Y|\mathbf{x}'}$ is invertible, since the c.d.f. of a unit Gaussian is invertible (though this function cannot be easily expressed).

Whereas in the simplified construction we showed that the neural network could approximate the inverse c.d.f., here we show that the second hidden layer of our larger Bayesian neural network can approximate the more complicated function required by lemma 3. This allows the second hidden layer of the neural network to transform $\mathbf{x}'$, $Z$ onto $\hat{\mathbf{Y}}$ such that $\hat{\mathbf{Y}}$ is appropriately similar to $\mathbf{Y}$. First we show that we can approximate the function $G_{\mathbf{x}'}^{-1}$, generating a probabilistic bound because the weights of the neural network are now Gaussian random variables:

**Lemma 4.** *For a uniformly continous function $G_{\mathbf{x}'}^{-1}(z) : z, \mathbf{x}' \mapsto \mathbf{y}$, for any $\epsilon, \delta > 0$ and compact subset $\mathcal{A}$ of $\mathbb{R}^D$, there exist fully-factorized Gaussian approximating distributions $q(\boldsymbol{\theta}_1)$, $q(\boldsymbol{\theta}_2)$, $q(\mathbf{b}_1)$, and $q(\mathbf{b}_2)$, and a function over the outputs of the later part of the neural network: $\hat{G}^{-1}(\mathbf{x}', z) \equiv \boldsymbol{\theta}_2(\sigma(\boldsymbol{\theta}_1 \mathbf{h}_1) + \mathbf{b}_1) + \mathbf{b}_2$ (remembering that $\mathbf{h}_1 \equiv \phi(z, \mathbf{x}')$), such that:*

$$\Pr\left( \left| \hat{G}^{-1}(\mathbf{x}', z) - G_{\mathbf{x}'}^{-1}(z) \right| > \epsilon \right) < \delta, \quad \forall \mathbf{x}' \in \mathcal{A}, z. \tag{45}$$

*The probability measure is over the weight distributions of $\boldsymbol{\theta}_1, \boldsymbol{\theta}_2, \mathbf{b}_1, \mathbf{b}_2$.*

Having shown that the second component can approximate the inverse c.d.f. to within a bound, we as before we further show that the random variable created by this transformation has a p.d.f. within a bound of the p.d.f. of $\mathbf{Y}$, which suffices to prove the desired result.

For this, we show this below for the transformed variable $\mathbf{x}'$:

**Lemma 5.** *For any $\epsilon > 0$ and $\delta > 0$ there exists a mean-field weight distribution $q(\boldsymbol{\theta}_1, \boldsymbol{\theta}_2, \mathbf{b}_1, \mathbf{b}_2)$ such that the probability density functions are bounded:*

$$\Pr\left( \left| p(y_i = \hat{\mathbf{Y}}_i) - p(y_i = \mathbf{Y}_i|\mathbf{x}', \mathcal{D}) \right| > \epsilon \right) < \delta, \quad \forall \mathbf{x}', y_i, \mathcal{A}. \tag{46}$$

We then move the bounds onto an expression in the original features, $\mathbf{x}$. Recall from before that because the variance of the weights in the first layer can be arbitrarily small, that for any $\epsilon$ there is a $\delta$:

$$\Pr\left( \|\mathbf{x}' - \mathbf{x}\| > \epsilon \right) < \delta \quad \forall \mathbf{x}, \mathbf{x}' \in \mathcal{A}, \tag{47}$$

where the probability measure is over $\boldsymbol{\theta}_{\mathrm{Pr}}$. Moreover, since we have assumed that the probability density function is continuous, this bound alongside the previous bound on the probability density functions jointly entail that:

$$\Pr\left( \left| p(y_i = \hat{\mathbf{Y}}_i) - p(y_i = \mathbf{Y}_i|\mathbf{x}, \mathcal{D}) \right| > \epsilon \right) < \delta, \quad \forall \mathbf{x} \in \mathcal{A}, y_i. \tag{48}$$

where the probability measure is over $\boldsymbol{\theta}_{\mathrm{Pr}}$.

$\square$

Below, we prove the lemmas required in the proposition above.

**Lemma 3.** *There exists continous function $G_{\mathbf{x}'}^{-1} = F_{Y|\mathbf{x}'}^{-1} \cdot F_Z \cdot \phi^{-1}$, where $F_Z$ is the c.d.f. of the unit Gaussian and $\phi^{-1}$ is the inverse of the activation function, such that the random variable $G_{\mathbf{x}'}^{-1}(\phi(Z))$ is equal in distribution to $\mathbf{Y}|\mathbf{x}$, if the p.d.f. of the posterior predictive is non-zero everywhere and the c.d.f. is continuous.*

*Proof.* Trivially $\phi^{-1}(\phi(Z)) = Z$.

Let $F_Z$ be the cummulative distribution function (c.d.f.) of the unit Gaussian random variable Z. By the Universality of the Uniform, $U = F_Z(Z)$ has a standard uniform distribution.

Let $F_{Y|\mathbf{x}}$ be the c.d.f. of $\mathbf{Y}$ conditioned on $\mathbf{x}$, and $\mathcal{D}$. Suppose that the posterior predictive is non-zero everywhere (that is, you cannot rule out that there's even the remotest chance of any $y_i$ given some input $\mathbf{x}$, however small). Then, since the c.d.f. is continuous by assumption, $F_{Y|\mathbf{x}}$ is invertible.

Again, by the Universality of the Uniform $U' = F_{Y|\mathbf{x}}(y)$ has a standard uniform distribution. So $\forall u : p(U = u) = p(U' = u)$. Moreover $F_{Y|\mathbf{x}}$ is invertible. So $p(Y = y) = F_{Y|\mathbf{x}}^{-1}(F_Z(Z))$.

It follows that there exists a continuous function as required.

$\square$

**Lemma 4.** *For a uniformly continous function $G_{\mathbf{x}'}^{-1}(z) : z, \mathbf{x}' \mapsto \mathbf{y}$, for any $\epsilon, \delta > 0$ and compact subset $\mathcal{A}$ of $\mathbb{R}^D$, there exist fully-factorized Gaussian approximating distributions $q(\boldsymbol{\theta}_1)$, $q(\boldsymbol{\theta}_2)$, $q(\mathbf{b}_1)$, and $q(\mathbf{b}_2)$, and a function over the outputs of the later part of the neural network: $\hat{G}^{-1}(\mathbf{x}', z) \equiv \boldsymbol{\theta}_2(\sigma(\boldsymbol{\theta}_1\mathbf{h}_1) + \mathbf{b}_1) + \mathbf{b}_2$ (remembering that $\mathbf{h}_1 \equiv \phi(z, \mathbf{x}')$), such that:*

$$\Pr\Big(\big|\hat{G}^{-1}(\mathbf{x}', z) - G_{\mathbf{x}'}^{-1}(z)\big| > \epsilon\Big) < \delta, \quad \forall \mathbf{x}' \in \mathcal{A}, z. \tag{45}$$

*The probability measure is over the weight distributions of $\boldsymbol{\theta}_1, \boldsymbol{\theta}_2, \mathbf{b}_1, \mathbf{b}_2$.*

*Proof.* The Universal Approximation Theorem (UAT) states that for any continuous function $f$, and an arbitrary fixed error, $e$, and compact subset $\mathcal{A}$ of $\mathbb{R}^D$, there exists a deterministic neural network with an arbitrarily wide single layer of hidden units and a non-polynomial activation, $\sigma$:

$$\forall \mathbf{x} \in \mathcal{A} : |\sigma(\mathbf{w}_2(\sigma(\mathbf{w}_1\mathbf{x}) + \mathbf{b}_1) + \mathbf{b}_2) - f(\mathbf{x})| < e. \tag{49}$$

In addition, we make use of Lemma 7 of [Foong et al., 2020]. This states that for any $e', \delta_2 > 0$, for some fixed means $\boldsymbol{\mu}_1, \boldsymbol{\mu}_2, \boldsymbol{\mu}_{b_1}, \boldsymbol{\mu}_{b_2}$ of $q(\boldsymbol{\theta}_1)$, $q(\boldsymbol{\theta}_2)$, $q(\mathbf{b}_1)$, and $q(\mathbf{b}_2)$ respectively, there exists some standard deviation $s' > 0$ for all those approximate posteriors such that for all $s < s'$, for any $\mathbf{h}_1 \equiv (z, \mathbf{x}') \in \mathbb{R}^{N+1}$

$$\Pr\Big(\big|\sigma(\boldsymbol{\theta}_2(\sigma(\boldsymbol{\theta}_1\mathbf{h}_1) + \mathbf{b}_1) + \mathbf{b}_2) - \sigma(\boldsymbol{\mu}_2(\sigma(\boldsymbol{\mu}_1\mathbf{h}_1) + \boldsymbol{\mu}_{b_1}) + \boldsymbol{\mu}_{b_2})\big| > e'\Big) < d. \tag{50}$$

Note that the deterministic weights of equation (49) can just be these means. As a result:

$$\Pr\Big(\big|\sigma(\boldsymbol{\theta}_2(\sigma(\boldsymbol{\theta}_1\mathbf{h}_1) + \mathbf{b}_1) + \mathbf{b}_2) - f(\mathbf{h}_1)\big| > e + e'\Big) < \delta. \tag{51}$$

We note that we define $\hat{G}^{-1}(\mathbf{x}', z) \equiv \sigma(\boldsymbol{\theta}_2(\sigma(\boldsymbol{\theta}_1\mathbf{h}_1) + \mathbf{b}_1) + \mathbf{b}_2)$ as above, and that $f(\mathbf{h}_1)$ may be $G_{\mathbf{x}'}^{-1}(z)$, which is assumed to be uniformly continuous. It follows, allowing $\epsilon = e + e'$:

$$\Pr\Big(\big|\hat{G}^{-1}(\mathbf{x}', z) - G_{\mathbf{x}'}^{-1}(z)\big| > \epsilon\Big) < \delta. \tag{52}$$

as required.

**Lemma 5.** *For any $\epsilon > 0$ and $\delta > 0$ there exists a mean-field weight distribution $q(\boldsymbol{\theta}_1, \boldsymbol{\theta}_2, \mathbf{b}_1, \mathbf{b}_2)$ such that the probability density functions are bounded:*

$$\Pr\Big(\big|p(y_i = \hat{\mathbf{Y}}_i) - p(y_i = \mathbf{Y}_i|\mathbf{x}', \mathcal{D})\big| > \epsilon\Big) < \delta, \quad \forall \mathbf{x}', y_i, \mathcal{A}. \tag{46}$$

In this lemma, we show that a bound on the inverse c.d.f. used to map $Z$ onto our target implies a bound in the p.d.f. of that constructed random variable to the p.d.f. of our target.

This lemma follows the argument of the simplified construction, with some additional complexity of notation introduced by the requirement that the input random variable was Gaussian rather than Uniform. Here, we complete the proof steps with a univariate $y$ to simplify notation, noting that because we can map $\mathbb{R} \to \mathbb{R}^K$ the multivariate regression follows trivially from the univariate result.

We first note that the result of Lemma 4 can be applied to $z' = \hat{G}(\mathbf{x}', y)$ using an inverted version of our network function such that:

$$\Pr\Big(\big|\hat{G}^{-1}(\mathbf{x}', \hat{G}(\mathbf{x}', y)) - G_{\mathbf{x}'}^{-1}(\hat{G}(\mathbf{x}', y))\big| > \epsilon_2\Big) < \delta_2, \quad \forall \mathbf{x}' \in \mathcal{A}, y. \tag{53}$$

We further note that by the triangle inequality:

$$\left| G_{\mathbf{x}'}^{-1}(G_{\mathbf{x}'}(y)) - G_{\mathbf{x}'}^{-1}(\hat{G}(\mathbf{x}', y)) \right| \leq \left| G_{\mathbf{x}'}^{-1}(G_{\mathbf{x}'}(y)) - \hat{G}^{-1}(\mathbf{x}', \hat{G}(\mathbf{x}', y)) \right| \tag{54}$$

$$+ \left| \hat{G}^{-1}(\mathbf{x}', \hat{G}(\mathbf{x}', y)) - G_{\mathbf{x}'}^{-1}(\hat{G}(\mathbf{x}', y)) \right|, \tag{55}$$

and since $G_{\mathbf{x}'}^{-1}(G_{\mathbf{x}'}(y)) = \hat{G}^{-1}(\mathbf{x}', \hat{G}(\mathbf{x}', y)) = \mathbf{x}'$:

$$\leq \left| \hat{G}^{-1}(\mathbf{x}', \hat{G}(\mathbf{x}', y)) - G_{\mathbf{x}'}^{-1}(\hat{G}(\mathbf{x}', y)) \right|. \tag{56}$$

Inserting this inequality into the result of Lemma 4, we have that:

$$\Pr\left( \left| G_{\mathbf{x}'}^{-1}(G_{\mathbf{x}'}(y)) - G_{\mathbf{x}'}^{-1}(\hat{G}(\mathbf{x}', y)) \right| > \epsilon_2 \right) < \delta_2, \quad \forall y, \mathbf{x}' \in \mathcal{A}. \tag{57}$$

But we further note that we have assumed that the c.d.f. $G_{\mathbf{x}'}$ is uniformly continous so for any $y'$ and $y''$, and for any $\epsilon' > 0$ there is an $\epsilon'' > 0$ and vice versa, such that if:

$$|y' - y''| < \epsilon', \tag{58}$$

then:

$$|G_{\mathbf{x}'}(y') - G_{\mathbf{x}'}(y'')| < \epsilon''. \tag{59}$$

It follows that for any $\epsilon$ and $\delta$ there is a $q(\boldsymbol{\theta})$ such that:

$$\Pr\left( \left| G_{\mathbf{x}'}(G_{\mathbf{x}'}^{-1}(G_{\mathbf{x}'}(y))) - G_{\mathbf{x}'}(G_{\mathbf{x}'}^{-1}(\hat{G}(\mathbf{x}', y))) \right| > \epsilon \right) < \delta, \quad \forall \mathbf{x}' \in \mathcal{A}, y \tag{60}$$

and therefore:

$$\Pr\left( \left| (G_{\mathbf{x}'}(y) - \hat{G}(\mathbf{x}', y) \right| > \epsilon \right) < \delta, \quad \forall \mathbf{x}' \in \mathcal{A}, y. \tag{61}$$

Next, we remember that $G_{\mathbf{x}'} = \phi \cdot F_Z^{-1} \cdot F_{Y|\mathbf{x}'}$, and that $\phi$ and $F_Z^{-1}$ are continuous, and therefore:

$$\Pr\left( \left| (F_{Y|\mathbf{x}'}(y) - F_{\hat{Y}|\mathbf{x}'}(y) \right| > \epsilon \right) < \delta, \quad \forall \mathbf{x}' \in \mathcal{A}, y, \tag{62}$$

where $F_{\hat{Y}|\mathbf{x}'}(y) = F_Z(\phi^{-1}(\hat{G}(\mathbf{x}', y)))$.

As a final step, we remember that the cumulative density is the integral of the probability density function. Therefore, by Theorem 7.17 of Rudin [1976] and the uniform convergence in the c.d.f.s, it follows that there for any bounds there exists $q(\boldsymbol{\theta})$ such that:

$$\Pr\left( \left| f_{\hat{Y}|\mathbf{x}'} - f_{Y|\mathbf{x}'} \right| > \epsilon \right) < \delta, \quad \forall \mathbf{x}' \in \mathcal{A}, y. \tag{63}$$

Writing out the probability density functions fully and mapping the univariate function the multivariate we have:

$$\Pr\left( \left| p(y_i = \hat{Y}_i) - p(y_i = Y_i | \mathbf{x}', \mathcal{D}) \right| > \epsilon \right) < \delta, \quad \forall \mathbf{x}' \in \mathcal{A}, y_i. \tag{64}$$

$\square$

## Footnotes

[4]`https://scikit-learn.org/stable/modules/generated/sklearn.datasets.make_moons.html#sklearn.datasets.make_moons`

[5]We set aside bias parameters, as they complicate the algebra, but adding them only strengthens the result because each bias term affects an entire row.

[6]Intuitively, we know that $\mathbf{x}^*$ is in all $\mathcal{A}_{\boldsymbol{\theta}_i}$, so when we add a new sample we know that there is either overlap around $\mathbf{x}^*$ or the point $\mathbf{x}^*$ is on the boundary of the new subset, which means we could equally well pick a different set that has $\mathbf{x}^*$ on its boundary and *does* have non-zero-measure overlap with the previous sets.

[7]A slightly complication is added by the continuity requirements. However, we note that the assumption that the p.d.f. is finite everywhere guarantees that there is a continuous function over $y$ which contains continous segments for each of $K$ dimensions, even if those individual segments are not continuous with each other.