[Reviews · NeurIPS 2020]

Review 1

Summary and Contributions: The paper demonstrates that the mean field assumption is not restrictive in deep neural networks. They analytically show that in some cases posterior predictive distributions can be estimated to arbitrarily small error with specific deep networks with mean field approximations. Experiments are provided to support the claim, suggesting no performance difference between mean-field and most structured priors.

Strengths: The paper's approach to the problem (How good/bad is mean field?) is solid, and the motivations, hypotheses, and background are clear. The authors provide direct analysis and experiments to support their hypotheses, and most assumptions needed for their theoretical contributions are reasonable. Most of the analysis is great and easy to follow, as well as directly supporting the claims in the main paper. The experiments generally well support the paper's claims. Each "hypothesis" is addressed directly with specific experiments.

Weaknesses: 1) The proof given for Theorem 2 is not sufficient. (see correctness) 2) Most of the results support the claim that there is not much gained by more complex weight posteriors, but I think the paper would benefit from an analysis of situations (or at least a simple one) where there -is- some difference between the mean-field and a more complex prior. Showing that such situations do not come up in practice (or say, diminish significantly as depth increases) would strengthen the motivation of the work. As it stands I'm not convinced that, empirically, anything novel is being observed. Most recent papers with BNNs have shown strong performance with mean field approximations, so it is not clear what new argument is being made or supported by the current set of results. 3) I have slight concerns regarding the paper's motivation. While I agree that there has been significant work towards finding ways to incorporate more complicated/structured approximate weight posteriors in BNNs, it's not completely clear to me that there is a wide-standing belief that mean field approximations are too restrictive in practice. I was not able to vet all references, but my understanding of a few, along with my experience with other recent BNN papers, does not suggest that there are practical concerns of insufficient approximation with mean field assumptions, rather more computational issues with respect to optimization.

Correctness: All other proofs appear correct to me. The proof of Theorem 2 is not correct/complete. The arguments made that "in practice it is very unlikely that a trainted neural network will turn off all its activations for any typical input..." is not a convincing argument. The authors agree that "because of...we cannot prove this analytically in general..." and then within the proof statement refer to experiments backing their claim. This should most likely be changed to a hypothesis/proposition, with clear notes in the main paper as to the how it is "mostly true in practice", or a more rigorous proof should be provided for "typical" data input settings. Code is provided and while I have not run it, there appears to be everything necessary to reproduce the empirical results of the paper.

Clarity: The paper is extremely well written. It is very easy to follow. Particularly, I was able to follow most if not all of the proofs provided in the appendices, and enjoyed the simplicity of some of their lines of attack with respect to proof techniques. Figures and tables are well done and effectively present the information the authors wanted readers to understand.

Relation to Prior Work: The authors provide ample references to support the claim that analysis specific to deep Bayesian neural networks has not addressed this issue with the depth provided in the paper. The most relevant and most inline-cited work, Foong et. al 2000, identify pathologies for single layer Relu-BNNs.

Reproducibility: Yes

Additional Feedback: The discussion does not contribute too much to the paper as it stands. It appears to simply be summarizing the entire paper, with language loosely similar to the introduction. This space could be better used to address drawbacks/limitations of the work, along with further research directions. I can see the potential value of this work to the community being as a strong step towards seriously understanding why mean-field assumptions are fine in practice with deep networks. I enjoy the analysis presented to support the use of mean-field approximations, but I think the central motivation that there is a "longstanding" belief that mean-field is restrictive is not really a prevailing opinion. The paper may benefit from a slight reframing, where the target of the paper is to support traditional assumptions made in practice, rather than challenging that those assumptions are too restrictive. Post Rebuttal: The other reviewer comments along with the authors' rebuttal have convinced me that my prior belief on what is "standard" is untrue, and underscores the papers' original argument that I previously critiqued. I am also satisfied with the authors' responses to all reviewer comments. However, reviewer discussion has led me to rethink the value of the experimental evaluation, and the weight it has for the paper's value. I think the simple simulation experiments described by the other reviewers may be necessary in a paper with this motivation and goal, without which a reader needs to trust that practically this plays out as described. I still believe this is a valuable contribution, but feel less confident in recommending acceptance.


Review 2

Summary and Contributions: The authors analyze the effects of MFVI on weight posteriors. In particular, the authors analyze whether mean field posteriors can represent richer posteriors in function space through adding depth than by utilizing correlated variational families. The authors postulate two hypotheses: 1. the weight distribution hypothesis, which trades of depth for covariance (roughly speaking) 2. the true posterior hypothesis, in which the authors postulate that for (almost) any BNN, there exists a mean field BNN with has the same predictive distribution. The authors evaluate both hypotheses with simple experiments and some theory.

Strengths: The authors pick a great topic, analyzing representation of the structure in BNNs is timely and interesting. In addition, I enjoyed the style of the paper clarifying its underlying hypotheses and trying to argue for them in particular, through experiment and theory. Both specific hypotheses are interesting to analyze.

Weaknesses: While I find the paper appealing, I am unsure about the results. First of all, the empirical results are unconvincing. In particular, the Figure showing in-between uncertainty is much less convincing than the published state of the art one can get there. there is way too little uncertainty to back up the claims that depth handles this issue. Further, the basic premise that depth induces correlation is well known and not particularly novel, and while I appreciate the authors' attempts to build solid arguments around specific hypotheses, the empirical analysis does not offer much backup. In particular the true posterior hypothesis is not backed up well here and not general enough. Also, for the weight distribution hypothesis, the authors restrict their arguments too much to the Gaussian case, while the strongest BNN approximations nowadays are non-Gaussian marginally. I also miss a very straightforward type of experiment, where the authors would simulate data and network weights with different structures and evaluate/visualize what a trained MFVI network with more depth/breadth could learn instead of only comparing learned models. Are they evaluating our ability to learn correlations with SGD or the structures of the weight models themselves? We can disambiguate those two sources of experimental uncertainty by simulating From known models with structure. The experiment comparing to HMC is particularly confusing and does not clearly support the second hypothesis. Why not focus on the predictive distributions here?

Correctness: The authors apply sound methodology, but their claims are somewhat restricted to special cases it seems. In particular, they focus on leaky Relus in order to be able to reason about the matrix products usefully. It is unclear to me at this point how generally useful these results are.

Clarity: The paper is professionally written, but occasionally not clear on why certain points are being made. I found myself losing track a few times, especially when trying to tie in the empirical content of the figures with the claims.

Relation to Prior Work: The authors ignore non-Gaussian VI. I do not find such an analysis useful without this field even discussed.

Reproducibility: No

Additional Feedback: Overall I found the paper confounding. I find that it contains a very strong idea, which is mainly the style in which it is written by specifying its hypotheses very clearly. Kudos to the authors, I now want to see this everywhere. But the actual message is not convincing or empirically lucid here and the theory feels contrived for simple cases. I hope the rebuttal will clarify some of my concerns about the main claims. Edit: Given the lack of clear positioning and weak demonstration of the types of models the authors suggest they can learn (i.e. in the toy example), and in particular the lack of simple simulations that would support the argumentations here, I will not improve my score. This is a pity, as I feel the paper could have followed a more convincing strategy to actually scientifically demonstrate the types of posteriors they argue for with convincing simulations in clear cases.


Review 3

Summary and Contributions: This paper tests out the effect of factorized posteriors in Bayesian deep neural networks. The authors demonstrate that complex correlations can emerge in factorized VI in DNNs, and then show that mean field VI can approximate the true function arbitrarily well in DNNs. Some experiments with popular approximate Bayesian inference methods are shown to demonstrate that mean field VI is not as bad as it often is made out to be. [Post-rebuttal: Thanks for the clarifications and especially the comparison to the Wasserstein distance. I've updated my score accordingly. With that being said, R2 makes some very good comments about the ability of MFVI to recover the true function _in practice_ rather than in theory that ought to be addressed. The alternative hypothesis for the results in this paper that still remains for me is that _deep MFVI_ works because it's _deep_ not because it's MFVI, and this ought to be considered a bit more in the camera ready.]

Strengths: Relevance: Bayesian deep learning and the understanding of methods for approximate Bayesian inference is pretty much a core part of the NeurIPS community, especially given the three tutorials in this and related spaces since NeurIPS 2018. Empirical evaluation: I'm most impressed by the experiments on mean field approximations whether variational or not using both SWAG and VOGN to demonstrate that the improvements over mean field / diagonal techniques can be marginal. Similarly, for a work that relies heavily on HMC on BNNs to demonstrate results, thank you for 1) showing convergence diagnostics (the acceptance rates all look reasonably sensible) and 2) showing the multi-modality of the parameters in Fig. 10. I also really like Fig 13 in the appendix as its a clean demonstration of the local hyper-plane product matrices that you are describing, and their local geometries. Significance: This work certainly challenges the assumptions made by many BDL researchers that "mean-field VI is poor performing from an algorithmic perspective and does not capture any correlations in function space." It certainly makes its case pretty well overall, and that is definitely a significant result for BDL researchers to consider when developing new approximate inference algorithms. Novelty: The experiments go well beyond the standard UCI, CIFAR, uncertainty quantification experiments and are genuinely interesting to think about, as well as genuinely novel. To progress in BDL, we probably need to devise better, more novel experiments, such as these to test if our approximate posteriors are doing what we think they are. Similarly, the universal approximation theorem and the weight correlation theorems approach their targets from a reasonably nice manner, and do so in a clever way. Theoretical grounding: The theory doesn't really seem to be much oversold and is mostly used to set up and be tested by the experiments. It looks reasonably well-written; I appreciate the notes in the Appendix about the strengths and weaknesses of Foong et al, 2020 as well as the in line note in line 875.

Weaknesses: Empirical evaluation: However, at least in the case of SWAG, the SGD/SWA baseline is also pretty close to the performance of both SWAG-Diag and SWAG in terms of both calibration, accuracy, and NLL. A similar case can be made for VOGN in comparison to Adam there. This seems to question the idea that it's really being Bayesian at all that matters -- if all methods end up performing about the same, rather than the mean field or non-mean field assumptions. A good baseline here would be to show SWA (no sampling) in Figure 5, and perhaps to display the plots as an improvement over a non-Bayesian approach. [This would be nice to show still and as you point out could help improving your approach.] I'm also not entirely sure of the extrapolation variance in Figure 1. Is there supposed to be a baseline here - in particular HMC? It looks like to get reasonable uncertainty, really even to cover the size of the data points, you have to move out four standard deviations. There's plenty of these types of problems in the literature, but for example: Fig 3 of https://arxiv.org/pdf/1907.07504.pdf shows a similar result here where the approximate Bayesian inference methods are doing better at interpolation and extrapolation variance than VI. [Calling depth VI "same magnitude" still feels like a bit of an overstatement, as the plot in the rebuttal definitely shows two-three x more variance in HMC.] A larger point, in comparison to the above paragraph is that the MFVI methods (and other diagonal counterparts) are still horrifically over-confident - as is shown in Figure 1. This really isn't addressed in the defense of MFVI in the paper, and is one of the main failings left open in the paper. For evidence of over-confidence, we only have to look at the ECEs reported for SWAG vs SWAG-Diag in Table 2, as well as Figure 5 of Foong et al, '20. While "modes of the true posterior ... may be approximately mean field," an alternative hypothesis could be that there are points? that fit well that can be captured by MFVI; however, these points have low surrounding area of the loss landscape and so are highly over-confident. By comparison, non-diagonal methods such as SWAG may be able to better capture this area and thus would have better uncertainty. [Indeed, it's suggestive that the reason why MFVI + NNs work is the NN part and all MFVI does is not mess the NN up too badly. I'm not particularly satisfied. But, this is probably an open research question in and of itself.] Significance and Relevance: The significance of the paper is a bit overstated as it's still not clear as to whether one would _want_ to use mean field VI, even if you can actually get some type of functional correlations out. There's long been anecdotal evidence that mean field VI is quite tricky to train in practice and doesn't perform well --- for a reference see https://arxiv.org/abs/2002.02405, https://arxiv.org/abs/1802.07044 and mentioned in Maddox et al, '19 along with all of the pitfalls mentioned in Foong et al, '20 as well --- for deep neural networks. This is suggestive that the work is defending something that much of the community has moved on from to some extent. [You end up being correct here due to the other reviewers.] Novelty: Much of the proofs are alluded to in Foong et al, '20. The primary extension is that mean field VI has the ability to produce a posterior predictive function that is consistent (in the sense of universal approximation), whereas the proof in Foong et al, '20 only shows that the mean and variance of functions are consistent. Theoretical grounding: I think that the universal approximation theorem as shown is a bit overblown, as Foong et al, '20 show that the first two moments will match (up to epsilon) in terms of worst case prediction to any function (which I believe should include the posterior predictive probabilities). Similarly, https://arxiv.org/abs/1908.04847 have shown that (generalized) sparse VI converges to the true function in a stronger sense -- their Appendix G allows for spike and slab mean field gaussian approximations, which is pretty close to MFVI plus dropout for example. I would suspect that their analysis is probably applicable to the spike and slab limit where the probability of the spike tends to zero, but that's somewhat of a guess. The other claim to theoretical novelty, demonstrating that linear product matrices (and likely non-linear even beyond piecewise local product functions) can exhibit correlations in the following layer is somewhat expected and indeed implied by some of the original work on MFVI, http://www.cs.toronto.edu/~fritz/absps/colt93.pdf, as we would certainly hope that the MFVI posterior still maintains the sorts of complex correlations that are necessary to model complex functions using NNs. [There's a distinction that needs to be drawn and ought to be made clear between "depth induces correlation" and "MFVI + depth induces _good_ correlation." With that being said, this is somewhat of a quibble, just be clear that you're not the only ones who have stated that depth induces correlation.]

Correctness: From a reasonable read of the technical material, it looks mostly correct with no overarching claims. I closely read the proof of Theorem 3 and it seems like it ought to hold. The restriction is to a single output function (no multiple outputs at the moment), but the comments make it clear that you could extend to >1 dimension.

Clarity: The key claim is that the induced product matrix, generated by matrix multiplying the weight posteriors without the activations can have non-linear correlations _even_ if it's mean field VI. This is relatively well described throughout but gets lost a bit in the weeds of the approximations and the bits on the universal approximation theorems. Story wise, this feels somewhat like a two-pronged defense of VI; however, in the experiments section, the two prongs could probably be somewhat better separated. In Section 6.2, it's a bit confusing for you to write q_{\full}(\theta) as that could easily correspond to a full-covariance variational posterior. The same applies to Fig. 4; with that being said you do have to approximate a Gaussian to those samples, and it is pretty clear that Fig. 4 is just measuring the distance between one mode and the MFVI Gaussian. This should probably be made more clear in the caption. One alternative is to test the similarity of the marginal distributions using a non-parametric method (e.g. Wasserstein) which would probably demonstrate that there's no real convergence to the true posterior - just to one of the posterior's modes. [Thank you.]

Relation to Prior Work: Overall, yes, the discussions between Foong et al, '20 and this paper are made very well distinguished. Honestly, it seems like a reasonably fair amount of interaction with the literature.

Reproducibility: Yes

Additional Feedback: In the rebuttal, I'd really like to see a better comparison to the HMC posterior in Figure 4, hopefully via products of Wasserstein distances. My concern is currently that the fit uni-modal Gaussians are overstating the amount of "convergence." [Thanks] I think that would help a lot with my concern that while "VI is able to fit an interpolating mode, it is an over-confident one" and to deal with my alternative loss surface based hypothesis on why VI can approximate the true function. That, and the concern about understating the collapse of variance from mean field VI, are the primary drivers in my "overall score" of the paper.


Review 4

Summary and Contributions: In this paper, the authors propose that performing full covariance posterior approximation in smaller networks is equivalent to mean field approximation in a deeper network, and suggest that as an alternative to complex full covariance approximation which is computationally costly. The paper states two hypotheses namely weight distribution hypothesis and true posterior hypothesis. Weight distribution hypothesis states that for any BNN with a full-covariance weight distribution, there exists a deeper BNN with a mean-field weight distribution that induces a “similar” posterior predictive distribution in function-space. True posterior hypothesis states that for any sufficiently deep and wide BNN, and for any posterior predictive, there exists a mean-field distribution over the weights of that BNN. The proved special cases of the hypothesis theoretically and demonstrated them in small data sets and deep neural networks.

Strengths: The study and observations made by the authors are significant and novel, and brings more clarity in mean field variational inference approximation for Bayesian neural networks. It challenges the common knowledge that mean field variational inference is restrictive. A very strong point is that they prove both theoretically and experimentally that mean field variational inference becomes less restrictive for deep neural networks and could capture the similar information captured by a full covariance approximation for shallow networks. They show that rich posterior approximations and deep architectures are complementary ways to create rich approximate posterior distributions over predictive functions. They proved the weight distribution hypothesis theoretically for linear networks and non linear networks with Relu activation. Matrix Variate Gaussian (MVG) distribution is shown to be a special case of the product matrix distribution formed out of mean-field weight matrices. True Posterior Hypothesis is proved using Universal Approximation Theorem for sufficiently wide networks. These hypothesis are supported empirically through various experiments on synthetic and real data sets.

Weaknesses: The observations made in the paper provide significant insights to mean field approximation in BNN and its usefullness. However, its doubtful if the study has some immediate practical implications as using mean field variational inference is a common practice in the community. Though mean field approximation is computationally faster, it comes at the cost of using a deeper and larger network as proposed by the authors through their weight distribution hypothesis. In practice, does this lead to a better computational and development time due to the use of a deeper network. This can further cause problems in model selection also. The theoretical proof of the weight distribution hypothesis are restricted to some special cases matrix-variate Gaussian and Relu activation, and is not shown to more general cases such as for activation functions other than Relu. Its also not clear the procedure to come up with a deep NN for a given full covariance approximation. Another weakness is that experimental setup does not effectively demonstrate the claims made in the weight distribution hypothesis.

Correctness: The theoretical proves about weight distribution hypothesis and true posterior hypothesis seems correct. However the experimental setup does not effectively demonstrate the claims made in the weight distribution hypothesis. The product matrix experiments in Figure 1 demonstrates weight distribution hypothesis to a good extend. However other experiments in section 6.1 fails to clearly demonstrate this. For uncertainty experiments it would have been good to also have the in between uncertainty predictions by a single layer full covariance neural network. The BNN used in Figure 3 is small and may not be appropriate to show existence of a deeper neural network for a shallow NN with FC approximation. Experiments in Figure 3 seem to demonstrate the true posterior hypothesis than the weight distribution hypothesis. In Figure 4, authors havent convincingly demonstrated the true distribution hypothesis that KL divergence goes to zero. In Figure 4 even with 6 layers, KL divergence is a large positive number. In Figure 3, its claimed that after two hidden layer MF catches FC, but that does not seem to be the case even though gap is smaller. Section 5 discusses about the true posterior hypothesis. The hypothesis is proved using universal approximation theorem which requires arbitrarily wide networks. How to choose the sufficient width required and the sufficient error in order to get low weight space ELBOs.

Clarity: The paper is well written and easily readable. However, it hasnt provided mathematical expressions and explanations which could have helped to understand the discussions better (Lemma 1 nd theorem 3). Instead it has been moved completely to the appendix. In Appendix D.1, the formation of co-variance function in equation 5 can be included. And notational details as to what small m in weight matrix and indices a,b,c,d correspond to can be added for more readability.

Relation to Prior Work: The paper discusses prior work and how this differs from prior works and contributions made in the paper.

Reproducibility: Yes

Additional Feedback: The paper is well-written and proof-read. Include mathematical expressions and explanations on various theorems. In Appendix D.1, the formation of co-variance function in equation 5 can be included. And notational details as to what small m in weight matrix and indices a,b,c,d correspond to can be added for more readability. Section 5 discusses about the true posterior hypothesis. The hypothesis is proved using universal approximation theorem which requires arbitrarily wide networks. How to choose the sufficient width required and the sufficient error in order to get low weight space ELBOs. Line 63, the acronym MFVI isn’t mentioned prior. More clarity on how the claim that even in narrow deep models there are modes of the true posterior that are approximately mean-field is made. ---------------------------AFTER REBUTTAL----------------------------- I thank the authors for addressing some of my concerns. However, it still does not address the immediate practical application of the study, as this leads to the requirement to have deeper and larger models.

[Author Response · NeurIPS 2020]

We want to express our gratitude to be part of a community that is so generous in offering such insightful and thorough
reviews. We were glad that reviewers thought that the paper was "extremely well written" with analysis that is "great
and easy to follow" and a "strong step towards seriously understanding why MF assumptions are fine in practice with
deep networks" [**R1**], "timely and interesting" [**R2**], with experiments that are "genuinely interesting to think about, as
well as genuinely novel" [**R3**], and "significant and novel" [**R4**].
[**R1**,**R3**,**R4**] **Motivation: R1** felt MFVI already shows "strong performance", excepting optimization problems, and
**R4** agreed its use was "common practice". In contrast, **R3** was unsure anyone "would want to use MFVI". 1) The
disagreement between reviewers shows how timely and relevant our work is. 2) This highlights how difficult it is to pin
down 'what everyone believes'. 3) To substantiate our view of what many published authors believe, our work cites 12
papers in §2 that say something like e.g. "a fully factorized posterior... is very restricting" [Louizos and Welling 2016.].
4) We agree about optimization issues, and discuss this on L109-110 as well as in S7. 5) We like the reframing **R1**
suggested, that our paper justifies common practice in addition to challenging a mistaken belief, and will add this. **But**
**whatever the reviewers personally believe, many publish based on the beliefs we critique, motivating our work.**
[**R2**,**R3**,**R4**] **Figure 2—in-between uncertainty**: This *toy* fig. shows that in-between uncertainty is possible in deeper
networks; larger-scale exps. should carry much more weight. But below we show that for 3-layer MFVI the uncertainty
is worse but of the same order-of-magnitude as an HMC baseline, while in 1-layer MFVI fails to capture uncertainty.
[**R2**,**R4**] **Special cases:** [**R2**,**R4**] **ReLUs**: Thm. 2 applies to all piecewise linear (PWL) activations (L160). This
includes ReLUs and many variants, which are very widely used. Many more theoretical papers focus on PWL, and
other nonlinearities can be approximated as PWL. [**R2**] **Gaussians**: in fact, lemmas 1 and 2 and Thm. 2 apply to any
distribution with finite first- and second- moments, which we will clarify. Also, fully-factored Gaussians are very
commonly used. [**R4**] **MVG**: We give the *general* form of the covariance (L125, D.1 eq. 15) and show equivalence to
the commonly used MVG distribution only as an intuitive lower-bound on the expressiveness of the covariance.
[**R2**,**R3**] **"Depth induces correlation is well-known"**: **R3** cites Hinton & van Camp 1993 and **R2** offers no citations.
H&vC *hypothesise* MF networks could learn complicated posterior distributions over functions, a two sentence comment
in their discussion (which we will cite). We *characterise* the form the correspondence takes and compare to the popular
MVG approximation. Moreover, many papers are motivated by the opposite view, see above.
[**R1**] **Completeness of Thm. 2**: This is a subtle point which we will make clearer by making Thm. 2 more mathemati-
cally precise. Thm. 2 is an existence proof ("can have non-zero covariance" L185) so proof by construction is valid
(L808). But it does not entail the WDH. We then make the admittedly informal argument based on experiment, which
you criticize, that the WDH represents a typical case. We can further develop the discussion of this in L189-191.
[**R1**] **Does full-covariance ever help?**: 1-layer FC beats MF. See below HMC or UCI in Louizos & Welling [2016].
[**R2**] **Correlations v. structure**: All exps. compare same model w/ and w/o correlations, which disambiguates this.
[**R3**] **MFVI over-confident?**: 1) Thanks for Wasserstein suggestion. See LHS fig. below comparing $W_2$ distance of FC
and MF Gaussian to HMC samples. The gap between $W_2$ lines is the distance cost imposed by using MF instead of FC.
The gap shrinks with depth, and is negligible after 4 layers, supporting our argument. **This directly addresses your**
**point and might justify an increased score**. 2) To clarify: our scope is defending MF assumption specifically, we
agree that VI and unimodal apprx. have problems which are not in our scope. 3) Table 2 ECE: As you say, for SWAG,
MF ECE is worse, but VOGN v. Noisy K-FAC tells opposite story. This makes sense if some but not all modes are
apprx. MF. VI finds the MF modes but SWAG training is indifferent to this (L310-313). Also ECE differences are small.
[**R3**] **Non-Bayesian baseline**: Fascinating point: note deterministic baseline has no correlations! Success of determin-
istic NNs supports our argument. We will add this. Note also a general defence of Bayesian methods is out of our scope.
[**R3**] **Novelty**: Only Thm. 3 draws from Foong et al. 2020, which we are very explicit about, while the rest of our
results were developed in parallel. Thm. 3 is only half a page of our paper, which makes it hard to see how this result is
"overblown". We will add discussion of Cherief-Abdellatif [2020], but observe (as you highlight) the setting considered
has significant differences. We feel that while you acknowledge the novelty of our experimental contribution, you might
also ascribe too much prescience to H&vC (see above) and have neglected key contributions (Thm.s 1 and 2).
[**R4**] **Exp. critique & Other**: 1) 1-layer HMC is below center. 2) Fig 3 sadly must be small (see C.3) but it does show
MF beat FC (maybe unclear in black-and-white, we will make fig. less color-dependent). 3) We discuss KL not going
to zero in L268. But also see Wasserstein fig. below LHS, where gap nearly disappears. 4) We agree deciding sufficient
width is an unsolved problem and list this and other problems in L214-221. 5) Thanks for the detailed further comments.

1-layer In-between Uncertainty

3-layer In-between Uncertainty

[Meta-Review · NeurIPS 2020]

The authors analyze the effects of MFVI on weight posteriors. In particular, the authors analyze whether mean field posteriors can represent richer posteriors in function space through adding depth than by utilizing correlated variational families. Strengths: -understanding the induced posterior correlations in function space is timely and interesting -most of the analysis is clear and easy to follow Weaknesses: - Theorems should be checked for rigorousness and be renamed as "propositions"